# CARE: Confounder-Aware Aggregation for Reliable LLM Evaluation

**Jitian Zhao** [* 1]   **Changho Shin** [* 2]   **Tzu-Heng Huang** [2]   **Satya Sai Srinath Namburi GNVV** [2]   **Frederic Sala** [2]

## Abstract

LLM-as-a-judge ensembles are the standard paradigm for scalable evaluation, but their aggregation mechanisms suffer from a fundamental flaw: they implicitly assume that judges provide independent estimates of true quality. However, in practice, LLM judges exhibit correlated errors caused by shared latent confounders—such as verbosity, stylistic preferences, or training artifacts—causing standard aggregation rules like majority vote or averaging to provide little gain or even amplify systematic mistakes. To address this, we introduce CARE, a confounder-aware aggregation framework that explicitly models LLM judge scores as arising from both a latent true-quality signal and shared confounding factors. Rather than heuristically re-weighting judges, CARE separates quality from confounders without access to ground-truth labels. We provide theoretical guarantees for identifiability and finite-sample recovery under shared confounders, and we quantify the systematic bias incurred when aggregation models omit confounding latent factors. Across 12 public benchmarks spanning continuous scoring, binary classification, and pairwise preference settings, CARE improves aggregation accuracy, reducing error by up to 26.8%.

## 1. Introduction

Large language models (LLMs) have become the workhorse solution for automated evaluation of model-generated outputs. *LLM-as-a-judge* systems offer a scalable alternative to expert annotation by substantially reducing cost and latency (Zheng et al., 2023a). Consequently, a common evaluation paradigm ensembles multiple LLM judges to pro-

duce consensus scores (Hu et al., 2024b). While effective, LLM-based evaluation outputs can be biased or only weakly aligned with human judgments. Moreover, these judges often exhibit correlated errors due to shared training data or modeling choices (Ye et al., 2025; Shi et al., 2024; Wang et al., 2024b; Deutsch et al., 2022; Li et al., 2025). As a result, heuristic aggregation strategies—such as majority vote, averaging, or ad-hoc reweighting—can yield diminishing returns and may even amplify shared mistakes.

Several techniques have been proposed to mitigate these issue, including *prompt-level interventions* and *specialized evaluators* (Chen et al., 2024; Li et al., 2024a; Zhu et al., 2025; Wang et al., 2024b). However, at the ensemble level, aggregation still typically defaults to majority voting or simple averaging (Li et al., 2024b), implicitly assuming that judges are *independent and similarly reliable*—assumptions that often fail in the presence of shared confounders and correlated errors. Additionally, existing approaches remain largely heuristic: they target specific failure modes or rely on assumptions that may be violated when judges share latent confounders. These limitations motivate the need for an aggregation framework that can ***explicitly model and account for shared confounders among judges***.

We propose CARE, a *confounder-aware aggregation* framework that models evaluations from multiple LLM judges through latent factors corresponding to both true quality and shared confounders. Rather than treating judges as independent sources of noisy labels, CARE explicitly separates the latent quality signal from spurious influences that simultaneously affect multiple judges. Specially, we instantiate CARE via two complementary estimators with different scopes:

- **CARE-SVD** leverages sparse-plus-low-rank structure in the judge score matrix to recover a dominant latent quality direction while accounting for shared confounders.
- **CARE-Tensor** builds on the learned dependency structure among judges to form an approximate multi-view representation, enabling identifiable recovery of quality and confounders through tensor decomposition.

We provide theoretical analysis establishing identifiability and finite-sample recovery guarantees for latent quality and confounders under shared confounding. These results characterize regimes in which heuristic aggregation meth-

[1] Department of Statistics, University of Wisconsin–Madison, Madison, WI, USA [2] Department of Computer Sciences, University of Wisconsin–Madison, Madison, WI, USA. Correspondence to: Jitian Zhao <jzhao326@wisc.edu>, Changho Shin <cshin23@wisc.edu>.

*Proceedings of the 43$^{rd}$ International Conference on Machine Learning*, Seoul, South Korea. PMLR 306, 2026. Copyright 2026 by the author(s).

ods are fundamentally misspecified and demonstrate when confounder-aware aggregation can reliably recover latent quality *without access to ground-truth labels*.

**Summary of Contributions.**

1. We introduce CARE, a confounder-aware framework for aggregating LLM-as-a-judge evaluations that explicitly models shared latent confounders among judges.
2. We develop two complementary estimators, CARE-SVD and CARE-Tensor, which operate under different information regimes and together support continuous, discrete, and preference-based evaluation settings.
3. We provide theoretical guarantees on identifiability and sample complexity in the presence of shared confounders, clarifying when reliable aggregation is possible.
4. We empirically demonstrate consistent improvements across diverse public benchmarks, ***achieving up to 26.8% error reduction*** over existing aggregation methods.

By explicitly modeling shared confounders during aggregation, CARE offers a principled alternative to heuristic ensembling and substantially improves the reliability of LLM-as-a-judge evaluation.

# 2. Background and Overview

We begin with brief background on automated evaluation using LLM judges and its connection to probabilistic graphical models.

**LLM-as-a-judge.** LLMs can act as inexpensive and fast proxies for human raters by assessing model generations via (i) *scalar quality scores* (e.g., 1–10 Likert or percentiles) (Zhu et al., 2025; Wang et al., 2024b; Shi et al., 2024), or (ii) *pairwise preferences* indicating which of two responses is better, as widely used in RLHF pipelines (Ouyang et al., 2022; Bai et al., 2022). Since individual LLM judges are often biased, recent work (Verga et al., 2024a) deploys *multiple* judges and aggregates their opinions—typically via majority vote or average pooling—to improve robustness. Our framework builds on this line of work ***but adopts a more principled approach to multi-judge aggregation by explicitly modeling shared confounders and correlated errors across judges***.

**Graphical Models and Latent-Variable MRFs.** Graphical models encode conditional independence structure in multivariate distributions, with Markov Random Fields (MRFs) being particularly well suited for modeling complex dependencies due to their flexibility, efficient inference, and amenability to structure learning. In the LLM-as-a-judge setting, we use MRFs to jointly model observed judge scores ($J$), shared confounding factors ($C$), and latent quality variables. This formulation allows us to capture structured dependencies among LLM evaluations while maintaining

computationally efficient and statistically tractable inference. When key influences are unobserved—most notably the true underlying quality of a model output—augmenting an MRF with latent nodes enables recovery of this hidden structure from noisy observations. This latent-variable MRF perspective is crucial in our context, offering a principled method to estimate the latent, true-quality signal from observable judges' scores while accounting for correlated judging errors.

# 3. CARE: Confounder-Aware Aggregation for Reliable Evaluation

We introduce CARE (Confounder-Aware Aggregation for Reliable Evaluation), a graphical model framework and algorithms that aggregates multiple judge scores into an estimate of latent quality while accounting for shared confounders.

## 3.1. Framework Setup

Let $X \in \mathbb{R}^{n \times p}$ be the judge-score matrix, where each row corresponds to an evaluation item and contains the $p$ judge scores. Let $H = (Q, C) \in \mathbb{R}^{k+1}$ collect a true-quality factor $Q$ and $k$ confounders $C$. Our graphical model encodes the conditional independence structure among the nodes in $(J, Q, C)$: if two nodes are not connected by an edge, they are independent conditioned on all other nodes. An example is shown on the right in Fig. 1. We assume this conditional dependence structure is *sparse*; i.e., there are not too many edges in the graph, and make this precise later on.

This framework is general and is compatible with a variety of distributions. For example, we may take $J, Q, C$ to involve discrete variables, Gaussians, or mixed. We can take the model to be an MRF or alternatively a mixture model. Our approaches are compatible with a broad range of choices, with practitioners able to select the most suitable modeling assumptions for their settings. In this paper, we focus on two concrete instantiations that match our experimental settings: CARE-SVD for continuous scores under a joint-Gaussian assumption, and CARE-Tensor for discrete or preference-based regimes where tensor methods are applicable.

**Goals and Assumptions.** Under the chosen modeling assumptions, our goal is to learn the distribution over $J, Q, C$. This involves handling ***three challenges***. First, **C1**: *we never observe the latents in $H$*—neither ground truth nor confounders. Second, **C2**: we *cannot assume any particular interaction* in the graph. Third, **C3**: even if we recover the model parameters, we must be able to distinguish between $Q$ and the confounders $C$ *to identify the model*. The latter is required to discover ***which latent is the ground-truth quality—and which is a confounder***. Once these obstacles are overcome, we seek to perform aggregation, e.g., com-

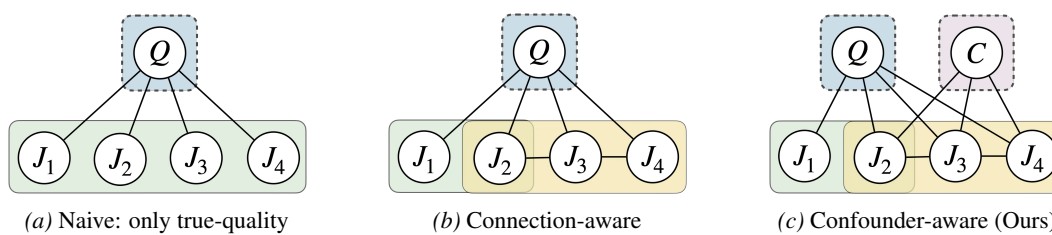

*Figure 1.* **Graphical models for aggregating judge scores under different structural assumptions. (a)** A naive model assumes scores reflect only a true latent quality ($Q$) and that all judges are equally reliable and represent independent views. **(b)** Connection-aware approach models intra-judge interactions ($J_2 - J_3 - J_4$), but still assumes the presence of a single latent quality score. **(c)** Our Confounder-aware model introduces additional latent confounders ($C$) influencing judge scores.

pute a posterior $P(Q|J)$, the Bayesian estimate for the latent true quality conditioned on all observable judge scores.

### 3.2. CARE Algorithm

The idea behind CARE is to examine two techniques, each of which is stymied by one of the obstacles **C2** or **C3** and to *delicately combine them in a novel way*. First, the sparsity of the conditional independence graph is encoded into a two-dimensional object that can be empirically estimated (e.g., the observable covariance matrix, or a cross-moment matrix). However, the presence of the latent variables (**C1**) obscures this structure—but a *sparse + low-rank decomposition* can reveal it (Chandrasekaran et al., 2012). However, while we can decompose the resulting low-rank term via SVD in the hope of identifying the model, we can only do so *up to rotations* and with additional assumptions. Therefore we are blocked by **C3**.

Conversely, tensor product decompositions (Anandkumar et al., 2014) exploit tensor rigidity to enable this decomposition to be uniquely identified. However, for these techniques the judges must be independent conditioned on the latents— and we cannot assume this by **C2**.

CARE (Algorithm 1) is a framework with a shared backbone and instantiations that combines these approaches. First, from the observed precision matrix $\hat{\Theta}(X)$ of judge scores, it estimates the underlying sparse graph structure via the sparse + low-rank decomposition which produces $\hat{S}$ that encodes judge-judge conditional dependency and $\hat{L}$ that encodes latent-judge dependency, overcoming **C1** and **C2**. Second, using the recovered sparse structure, CARE selects subsets of judges with sufficient conditional independence, constructs a higher-order moment that can be decomposed to recover latent factors, mitigating **C3**.

The remaining *label* ambiguity (**C3**) for deciding which recovered component corresponds to the true-quality factor $Q$ rather than a confounder, can be resolved by a lightweight symmetry-breaking step. This requires a weak assumption on the quality of the judges. In practice, even this assumption can be removed by employing simple heuristics to identify the true-quality factor among the latent factors. Fi-

nally, we aggregate judge scores into robust evaluations by weighting according to loadings from the identified quality factor.

CARE provides a shared backbone with model-specific *instantiations*. CARE-SVD uses only second-order structure and a symmetry-breaking rule to select the quality factor, while CARE-Tensor uses the sparse dependency structure to form approximately independent judge groups ("views") and then applies tensor decomposition for identifiable recovery in discrete/mixture regimes. Next, we study two special cases that instantiate this template for different data regimes (fully Gaussian vs. mixture settings); more general settings are shown in the Appendix.

**CARE for Gaussian Mixtures.** For illustrative purposes, we demonstrate following case with one binary confounder and one binary true quality variable. However, this can be easily extended to multi-class or multiple confounders through our framework. We refer this method as CARE-Tensor.

Consider a Gaussian mixture with binary latent variables $(Q, C) \in \{0, 1\}^2$ and mixing weights $\pi_{qc}$, where

$$J \mid (Q = q, C = c) \sim \mathcal{N}(\mu_{qc}, \Sigma), \qquad (q, c) \in \{0, 1\}^2.$$

The sparse + low-rank decomposition produces $(\hat{S}, \hat{L})$; the learned sparse graph $\hat{S}$ induces a partition of judges into three conditionally independent groups $X_1, X_2, X_3$. For such a partition, third-order cross-moments factorize:

$$\mathbb{E}\big[X_1 \otimes X_2 \otimes X_3 \,\big|\, Q, C\big]$$
$$= \mathbb{E}[X_1 \mid Q, C] \otimes \mathbb{E}[X_2 \mid Q, C] \otimes \mathbb{E}[X_3 \mid Q, C].$$

Consequently, the population tensor decomposes as

$$T := \mathbb{E}[X_1 \otimes X_2 \otimes X_3] = \sum_{q,c} \pi_{qc} \mu_{qc}^{(1)} \otimes \mu_{qc}^{(2)} \otimes \mu_{qc}^{(3)},$$

where $\mu_{qc}^{(k)}$ denotes the mean restricted to group $X_k$. A CP decomposition of $T$ recovers $\{\mu_{qc}^{(k)}\}$, concatenating the groups yields $\{\mu_{qc}\}$, and the weights $\{\pi_{qc}\}$ follow from

component norms. With $(\mu_{qc}, \Sigma, \pi_{qc})$, Bayes' rule gives the posteriors

$$\alpha_{qc}(J) := \Pr(Q = q, C = c \mid J) \propto \pi_{qc}\, \phi(J; \mu_{qc}, \Sigma),$$
$$\mathbb{E}[Q \mid J] = \sum_{c \in \{0,1\}} \alpha_{1c}(J). \tag{1}$$

where $\phi(\cdot; \mu, \Sigma)$ is the Gaussian density. Thus, in the mixed regime CARE leverages third-order moments (enabled by $\hat{S}$) to identify the conditional means and mixture proportions.

This setup also applies when each judge emits a binary label $X_k \in \{0,1\}^{d_k}$. The sparse-plus-low-rank decomposition and the induced three-view partition are unchanged, and the third-order moment factorization holds with $\mu_{qc}^{(k)} := \mathbb{E}[X_k \mid Q{=}q, C{=}c] \in [0,1]^{d_k}$ interpreted as per-judge success probabilities. A CP decomposition of $T$ recovers $\{\mu_{qc}^{(k)}\}$ and $\{\pi_{qc}\}$ exactly as in the Gaussian case. For the final posterior we adopt a *Gaussian prior* $\alpha_{qc}(J) \propto \pi_{qc}\, \phi(J; \mu_{qc}, \Sigma)$ to preserve residual within-view dependence among judges; this leaves the identifiable tensor backbone intact and performs well in practice. In Appendix C, we also cover the fully binary setting, where judges emit binary labels and $(Q, C)$ are binary.

**CARE for Fully Gaussian.** When all variables are jointly Gaussian, odd central moments vanish, so third-order tensors carry no information. CARE therefore relies on second-order structure, which we refer this method as CARE-SVD. Let $\Sigma = \mathrm{Cov}\big[(J, H)^\top\big]$, $K = \Sigma^{-1} = \begin{pmatrix} K_{JJ} & K_{JH} \\ K_{HJ} & K_{HH} \end{pmatrix}$. The marginal precision of $J$ satisfies

$$\Theta := \Sigma_{JJ}^{-1} = K_{JJ} - K_{JH} K_{HH}^{-1} K_{HJ} = S - L,$$

with sparse component $S := K_{JJ}$ and low-rank component $L := K_{JH} K_{HH}^{-1} K_{HJ}$. Under the identifiability conditions summarized in Section 4, the eigenspace of $\hat{L}$ recovers latent–judge loadings up to sign and permutation, and we obtain an estimate of the latent quality by aggregating the judges with weights from the identified "quality" factor.

### 3.3. Heuristics for Identifiability and Robust Estimation

Any instantiation of CARE will require symmetry-breaking procedures for latent variable identifiability. For example, the fully Gaussian case needs a heuristic to identify the true-quality direction among latent factors, distinguishing $Q$ from confounders $C$. In the binary-Gaussian mixture scenario, an additional step resolves ambiguity between latent states ($Q = 0$ vs. $Q = 1$). Doing so will require additional information that can come from modeling assumptions, the use of ground-truth samples, or heuristics. We detail some examples below:

**Identifying True-Quality Factor for Fully Gaussian Model.** Symmetry breaking in the fully Gaussian setting

---

**Algorithm 1** CARE: Confounder-Aware Aggregation for Reliable Evaluation

---

**Require:** Judge score matrix $X \in \mathbb{R}^{n \times p}$, parameters $(\gamma_n, \tau)$, decomposition method $\mathcal{D} \in \{\mathrm{SVD}, \mathrm{Tensor}\}$
**Ensure:** Estimated True Quality $\{\hat{q}^{(i)}\}_{i=1}^n$

1: **Graph Sparse Structure Estimation:** Compute precision of the observed matrix $\hat{\Theta}(X)$.

2: **Sparse + low-rank decomposition:**

$$(\hat{S}, \hat{L}) \in \arg\min_{S,L} \frac{1}{2} \big\| \widehat{\Theta}(X) - (S - L) \big\|_F^2$$
$$+ \gamma_n (\|S\|_1 + \tau \|L\|_*).$$

3: **Latent Factor Extraction:**
4: **if** $\mathcal{D} = \mathrm{SVD}$ **then** {Fully Gaussian}
5:     Compute $U \Lambda U^\top \leftarrow \mathrm{SVD}(\hat{L})$, where $U \in \mathbb{R}^{p \times h}$
6: **else if** $\mathcal{D} = \mathrm{Tensor}$ **then** {Gaussian mixture}
7:     Partition judges into conditionally independent groups with $\hat{S}$
8:     Form empirical third-order tensor from judge groups
9:     Run tensor decomposition, obtain latent conditional means $\mu_{qc}$ and mixture proportions $\pi_{qc}$
10: **end if**
11: **Symmetry Breaking:** Identify the true-quality factor using heuristics described in §3.3
12: **Latent Quality Estimation:** Use the identified quality factor, compute $\hat{q}^{(i)}$ for each example, where $\hat{q}^{(i)} = P(Q = 1 \mid J_i)$ for mixture model or $\hat{q}^{(i)} = \mathbb{E}[Q \mid J]$ for fully gaussian. See formulas for each instantiation.

---

amounts to selecting which recovered low-rank direction corresponds to the true-quality latent $Q$ (rather than a confounder). Our guiding principle is that quality induces the strongest *shared* variation across judges, whereas confounders are typically weaker or concentrated on a subset. Accordingly, we take the leading eigenvector of the fitted low-rank component $\hat{L}$ as the shared "quality" axis and treat its associated factor as $Q$. This choice is objective and works well when the dominant shared variation reflects quality. When this assumption may be violated, we use lightweight alternatives (Appendix C): (a) a small human-rated anchor set, or (b) a balanced-loading criterion that favors broadly distributed loadings. We also validate the leading-eigenvector rule empirically in Appendix E.

**Identifying Latent States for Mixed Model.** After recovering component means $\{\hat{\mu}_r\}$, we compute the leading eigenvector $v$ of $\hat{L}$ and score states by $s_r = v^\top \hat{\mu}_r$; the highest-scoring state(s) are mapped to $Q{=}1$ and the remainder to $Q{=}0$. If labeled anchors are available, we optionally verify this mapping by choosing the state whose means best match the $Q{=}1$ labeled subset (details in Appendix C).

# 4. Theoretical Analysis

We formalize when CARE recovers the quality/confounder structure and how much data is needed. The section proceeds in four steps: (i) *exact identifiability* under clean structure; (ii) *stability* under perturbations; (iii) *finite-sample* rates for the joint-Gaussian (spectral) path; and (iv) *sample complexity* for the binary Gaussian mixture (tensor) path. We conclude with a brief remark on misspecification. All technical assumptions and proofs are deferred to the Appendix.

**Notation and setup.** Let $J \in \mathbb{R}^p$ denote judge scores. We write the observable precision as $K_{JJ} = S - L$, where $S$ is sparse and $L$ is low rank. In the joint-Gaussian setup, $L = K_{JH} K_{HH}^{-1} K_{HJ}$ with latent factors $H \in \mathbb{R}^h$. In the mixed binary Gaussian with three conditionally independent judge groups, we exploit a factorized third-order moment built from the three views.

We start with fully Gaussian case. To ensure that latent directions are recoverable from $L$, we assume latent independence and orthogonality in the latent-observable link. Under these conditions, the columns of $K_{JH}$ (and hence the eigen-directions of $L$) are uniquely identified up to sign and permutation; moreover, the recovery is stable to small violations of orthogonality.

**Proposition 4.1** (Identifiability and stability of latent—judge directions)**.** *Assume $K_{HH} = \text{diag}(d_1, \ldots, d_h)$ with $d_1 > \cdots > d_h > 0$ and the columns of $K_{JH}$ are orthogonal. Then the columns of $K_{JH}$ (equivalently, the latent directions encoded by $L$) are identifiable from $L$ up to sign and permutation. Moreover, if $K_{JH}$ is perturbed to $\tilde{K}_{JH} = K_{JH} + E$, letting $\delta_i$ denote the eigengap of $L$ at eigenvector $u_i$, then*

$$\|\hat{u}_i - u_i\|_2 \lesssim \|K_{HH}^{-1}\|_2 \frac{\|E\|_2}{\delta_i} \qquad (i \in [h]).$$

We next quantify how many samples suffice to estimate these directions from data via $S+L$ estimation, highlighting the role of eigengaps and the curvature of the low-rank tangent space.

**Theorem 4.2** (Finite-sample recovery for the spectral path)**.** *Let $L^\star = K_{JH} K_{HH}^{-1} K_{HJ}$ have global eigengap $\delta > 0$, and let $\xi(T)$ denote the curvature constant of the tangent space $T(L^\star)$ governing $S+L$ estimation. Under standard identifiability/incoherence/curvature conditions (Appendix D), for any $\eta > 0$, with probability at least $1 - 2e^{-\eta}$,*

$$\max_{i \leq h} \|\hat{u}_i - u_i\|_2 = O\left(\sqrt{\frac{\eta}{n}} \frac{1}{\xi(T)\,\delta}\right).$$

For the mixed binary Gaussian model with three conditionally independent judge groups, we show that the CP (CAN-

DECOMP/PARAFAC) structure of the third-order cross-moment enables recovery of mixture means and weights at a rate that scales with noise, separation (gap), and the number of judges.

**Theorem 4.3** (Sample complexity for recovering $(\mu_{qc}, \pi_{qc})$)**.** *Assume the three judge groups are conditionally independent given $(Q, C) \in \{0, 1\}^2$, the third-order moment admits a CP decomposition with spectral gap $\delta > 0$, and the within-component observation covariance $\Sigma$ satisfies $\|\Sigma\|_{\text{op}} \leq \sigma_{\max}^2$. Let $p$ be the total number of judges and $\pi_{\min} = \min_{q,c} \pi_{qc}$. If*

$$n \gtrsim \frac{\sigma_{\max}^6}{\delta^2 \, \pi_{\min}^2} p \, \log(p/\varepsilon),$$

*then with probability at least $1 - \varepsilon$,*

$$\max_{q,c} \|\hat{\mu}_{qc} - \mu_{qc}\|_2 \lesssim \frac{\sigma_{\max}^3}{\delta} \sqrt{\frac{p \log(p/\varepsilon)}{n}},$$

$$\max_{q,c} |\hat{\pi}_{qc} - \pi_{qc}| \lesssim \sqrt{\frac{p \log(p/\varepsilon)}{n}}.$$

Finally, when the model omits confounding latent factors, the estimated conditional means incur a systematic bias. We provide an explicit bound on this misspecification error in Appendix D, quantifying how it scales with the strength of the omitted confounders and their alignment with the recovered directions.

# 5. Experimental Results

We evaluate the effectiveness of CARE across diverse experimental setups, real-world and semi-synthetic scenarios. Our goal is to validate the following key claims:

- **Improving aggregation of LLM judges:** CARE produces more accurate and robust aggregate scores from multiple LLM judges compared to baselines (Section 5.1).
- **Interpreting latent confounding factors:** CARE's latent structure enables diagnosis by revealing associations between latent confounders and observable confounder attributes (e.g., length/style/readability) (Section 5.2).
- **Effective Integration of Program Judges:** CARE integrates programmatic judges, known to have high bias, by explicitly modeling their biases (Section 5.3).
- **Demonstrating Robustness under Controlled Confounding Factors:** CARE remains accurate when evaluations are deliberately affected by controlled biases, as demonstrated by the semi-synthetic data from (Chen et al., 2024) (Section 5.4).
- **Effective Defense against Adversarial Answers.** Zhao et al. (2025) shows that inserting only a few tokens can fool LLM judges into producing incorrect evaluations. Building on such setup, we show that CARE provides an effective defense against such attacks (Section 5.5).

*Table 1.* Aggregation performance across different scoring datasets, measured by MAE (↓; mean ± std).

|  | ASSET | FeedbackQA | Review-5K | Summarize | UltraFeedback | Yelp |
|---|---|---|---|---|---|---|
| MV | $31.153 \pm 0.000$ | $0.822 \pm 0.000$ | $2.608 \pm 0.000$ | $1.417 \pm 0.000$ | $0.851 \pm 0.000$ | $0.923 \pm 0.000$ |
| AVG | $33.663 \pm 0.000$ | $0.830 \pm 0.000$ | $2.274 \pm 0.000$ | $1.394 \pm 0.000$ | $0.686 \pm 0.000$ | $1.037 \pm 0.000$ |
| WS | $29.073 \pm 0.436$ | $0.793 \pm 0.009$ | $2.593 \pm 0.052$ | $1.364 \pm 0.007$ | $0.829 \pm 0.009$ | $0.977 \pm 0.008$ |
| UWS | $33.928 \pm 0.000$ | $0.875 \pm 0.000$ | $2.602 \pm 0.000$ | $1.362 \pm 0.000$ | $0.680 \pm 0.000$ | $0.987 \pm 0.000$ |
| CARE-SVD | $\mathbf{27.629 \pm 0.156}$ | $\mathbf{0.730 \pm 0.002}$ | $\mathbf{1.957 \pm 0.018}$ | $\mathbf{1.325 \pm 0.004}$ | $\mathbf{0.623 \pm 0.006}$ | $\mathbf{0.694 \pm 0.004}$ |

*Table 2.* Aggregation performance across classification and preference datasets, measured by Accuracy (↑).

|  | Chatbot-Arena | CivilComments | PKU-BETTER | PKU-SAFER | SHP | Summarize |
|---|---|---|---|---|---|---|
| MV | $0.517 \pm 0.000$ | $0.691 \pm 0.000$ | $0.701 \pm 0.000$ | $0.698 \pm 0.000$ | $0.626 \pm 0.000$ | $0.600 \pm 0.000$ |
| AVG | $0.551 \pm 0.000$ | $0.690 \pm 0.000$ | $0.726 \pm 0.000$ | $0.717 \pm 0.000$ | $0.634 \pm 0.000$ | $0.683 \pm 0.000$ |
| WS | $0.543 \pm 0.004$ | $0.739 \pm 0.003$ | $0.575 \pm 0.002$ | $0.570 \pm 0.001$ | $0.619 \pm 0.005$ | $0.705 \pm 0.002$ |
| UWS | $0.507 \pm 0.000$ | $0.713 \pm 0.000$ | $0.703 \pm 0.000$ | $0.701 \pm 0.000$ | $0.629 \pm 0.000$ | $0.713 \pm 0.000$ |
| Dawid–Skene | $0.546 \pm 0.000$ | $0.735 \pm 0.000$ | $0.551 \pm 0.000$ | $0.548 \pm 0.000$ | $0.612 \pm 0.000$ | $0.705 \pm 0.000$ |
| GLAD | $0.510 \pm 0.009$ | $0.695 \pm 0.002$ | $0.697 \pm 0.008$ | $0.671 \pm 0.013$ | $0.644 \pm 0.000$ | $0.718 \pm 0.013$ |
| MACE | $0.550 \pm 0.000$ | $0.732 \pm 0.000$ | $0.734 \pm 0.000$ | $\mathbf{0.735 \pm 0.000}$ | $0.580 \pm 0.000$ | $0.706 \pm 0.000$ |
| CARE-SVD | $\mathbf{0.580 \pm 0.004}$ | $\mathbf{0.778 \pm 0.004}$ | $0.691 \pm 0.002$ | $0.690 \pm 0.002$ | $0.543 \pm 0.006$ | $0.695 \pm 0.003$ |
| CARE-Tensor | $0.564 \pm 0.013$ | $0.749 \pm 0.002$ | $\mathbf{0.779 \pm 0.002}$ | $0.731 \pm 0.002$ | $\mathbf{0.695 \pm 0.008}$ | $\mathbf{0.814 \pm 0.001}$ |

**Datasets & Metrics.** We evaluate on 12 public benchmarks covering continuous scoring (QA/feedback, summarization, reviews) and binary/pairwise classification (toxicity and preference). We report MAE for scoring datasets and Accuracy for classification/preference datasets; the full dataset list is in Appendix E.

**Baselines.** We compare CARE to following baseline aggregation methods: (i) majority voting (MV), (ii) simple averaging (AVG) (Li et al., 2024b), (iii) discrete-based weak supervision (WS) (Bach et al., 2019a), and (iv) continuous-based weak supervision (UWS) (Shin et al., 2022). Both WS and UWS are probabilistic graphical models that do not account for confounding factors (WS operates on discrete labels, while UWS handles continuous scores). In the Gaussian mixture setting, where quality factors are treated as discrete, we additionally include classical aggregation baselines: Dawid–Skene (Dawid & Skene, 1979), GLAD (Whitehill et al., 2009), and MACE (Hovy et al., 2013).

**LLM Judges.** We use 11 - 20 LLMs as our judges ranging from 0.6B to 14B parameters and extract their judging scores. The full model list is provided in Appendix E. In scoring tasks, LLM judges assign numeric ratings within ranges aligned to human annotations. In classification tasks, they output scores from 0–9, with 4.5 serving as a threshold for binary decisions, while in preference tasks they assign values from –3 to 3 to indicate the relative strength of preference.

### 5.1. Improving Aggregation of LLM judges

**Setup.** We compare aggregation methods using CARE-SVD and CARE-Tensor. To ensure consistency, we adapt the prompt template from (Roucher, 2025), modifying it to fit our experimental setup. The exact prompt used is

provided in Appendix E.

**Results.** Table 1 and Table 2 show that CARE consistently outperforms baseline methods across both scoring and classification/preference tasks. On scoring datasets, CARE-SVD achieves the lowest MAE in all cases, reducing error by up to 26.8% compared to MV on UltraFeedback. Averaged across scoring datasets, CARE-SVD yields a 17.37% relative improvement over AVG and a 12.75% improvement over MV. On classification and preference benchmarks, CARE achieves the best accuracy on 5 of 6 datasets, with CARE-Tensor leading on three (PKU-BETTER, SHP, and Summarize), including a 13.4% relative improvement in accuracy on Summarize over the strongest baseline. Overall, these results highlight CARE's ability to reduce compounding biases and deliver more reliable aggregation across diverse evaluation settings.

### 5.2. Interpreting Latent Confounding Factors

CARE can also be used as a diagnostic tool. Beyond producing an aggregate score, its latent factors can point to response attributes that systematically affect judges but are not necessarily part of the underlying quality signal. We illustrate this with one representative example for each setting. The basic procedure is simple: we first convert each learned latent component into an example-level score, and then compare these scores with observable response features such as length, formatting, and readability. These correlations are meant to help interpret the learned factors; we do not claim that they provide a unique causal explanation for the factor.

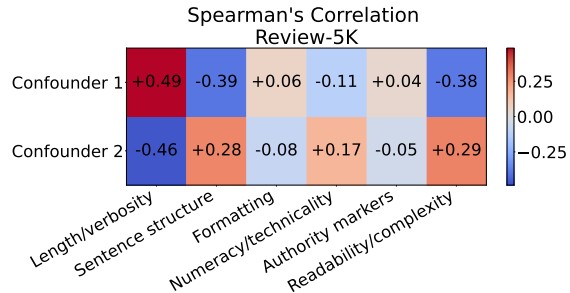

*Figure 2.* Interpreting CARE-SVD latent confounders on Review-5K. Heatmap reports Spearman correlations between inferred confounder scores and response features.

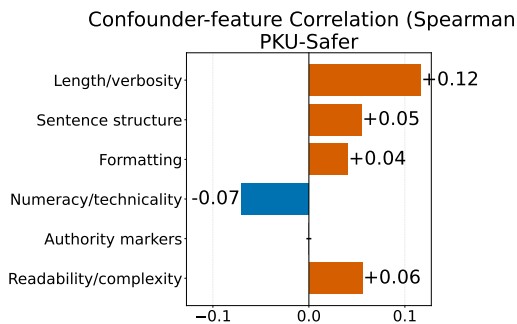

*Figure 3.* Interpreting CARE-Tensor latent confounders on PKU-Safer. Bars show Spearman correlations between inferred confounder posteriors and response features.

### 5.2.1. CARE-SVD (FULLY GAUSSIAN)

**Setup.** For CARE-SVD, we use the leading eigen-directions of the learned low-rank component $\hat{L}$. For each example $i$ and latent direction $u_r$, we compute a factor score $z_r(i) = \langle u_r, J_i \rangle$. We then correlate each non-quality factor score with a small set of response-level features that capture common sources of judge bias:

- **Length/verbosity**: total character length of the response.
- **Sentence structure**: average sentence length.
- **Formatting**: number of numbered-list items.
- **Numeracy/technicality**: amount of number-heavy content after regressing out length.
- **Authority markers**: citation-like markers, such as bracketed references or URLs.
- **Readability/complexity**: SMOG score, an estimated grade level based on words with three or more syllables.

**Results.** Figure 2 shows the correlations on Review-5K, the ICLR 2024 peer-review dataset. The first confounder is most associated with longer reviews: it has a positive correlation with response length ($\rho \approx 0.49$), and negative correlations with average sentence length and SMOG complexity ($\rho \approx -0.39$ and $-0.38$). The second confounder follows the opposite pattern. It is associated with shorter reviews ($\rho \approx -0.46$), but also with more complex sentence structure and higher readability complexity ($\rho \approx 0.28$–$0.29$), as well as slightly more numeric content ($\rho \approx 0.17$). These patterns are consistent with plausible peer-review confounders: judges may respond to surface cues such as verbosity, technical density, or writing style, even when these cues are not the same as paper quality.

### 5.2.2. CARE-TENSOR (GAUSSIAN MIXTURE)

**Setup.** In the CARE-Tensor setting, interpretability is more direct. We use the posterior confounder probability $\Pr(C = 1 \mid J)$ as an example-level confounder score and examine its association with human-interpretable response attributes. We use the same set of concepts as in the fully Gaussian case and report results on PKU-Safer.

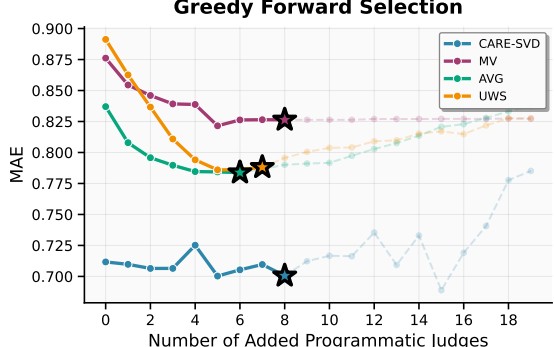

*Figure 4.* Integration results on FeedbackQA dataset with greedy judge selection guided by validation dataset performance.

**Results.** Figure 3 shows that the inferred confounder is most strongly associated with verbosity-related attributes. Length/verbosity exhibits the largest positive correlation with $\Pr(C = 1 \mid J)$ ($\rho \approx +0.12$), while numeracy/technicality shows a modest negative association ($\rho \approx -0.07$).

### 5.3. Effective Integration of Programmatic Judges

**Setup.** Next, we evaluate CARE by integrating it with 30 synthesized programmatic judges. Each judge is constructed by prompting OpenAI's GPT-4o to generate evaluation logic, which is then compiled into an executable program (Huang et al., 2025). Programmatic judges are designed to assess responses along specific dimensions such as *structure, readability, safety, relevance, and factuality*. While cost-effective, their deterministic nature can introduce *systematic biases*, resulting in noisy evaluation signals. This setup creates a challenging aggregation scenario to test CARE's robustness. Details of the judge generation process are in Appendix E.

We progressively integrate programmatic judges in a greedy manner: at each step, we add a judge that yields the largest improvement on a validation dataset (10% of training dataset), measured by MAE. The process stops when no further MAE reduction occurs for four consecutive steps. We

*Table 3.* Robustness to artificially injected bias. Reported as MAE between bias-injected and original versions; lower values indicate stronger robustness.

|  | Beauty Bias | Authority Bias |
|---|---|---|
| MV | 0.919 | 0.824 |
| AVG | 0.506 | 0.325 |
| WS | 1.229 | 0.754 |
| UWS | 0.508 | 0.271 |
| CARE-SVD | **0.375** | **0.233** |

*Table 4.* False positive rates on adversarial responses. CARE-Tensor can reduce false positives significantly across cases.

| Attack type | MV | WS | CARE-Tensor |
|---|---|---|---|
| ":" | 0.577 | 0.520 | **0.496** |
| "," | 0.441 | 0.546 | **0.000** |
| "Solution" | 0.558 | 0.522 | **0.371** |
| "Thought process:" | 0.674 | 0.587 | **0.000** |
| " " | 0.641 | 0.577 | **0.432** |
| "Step by step" | 0.571 | 0.504 | **0.392** |

then evaluate aggregation methods on the FEEDBACKQA dataset, consistent with previous experiments.

**Results.** Figure 4 shows aggregation performance as judges are progressively added. With the early-stopping criterion, CARE achieves the lowest error by integrating eight additional programmatic judges, reducing MAE by 15.2% compared to MV and by 11.1% compared to UWS. These results highlight CARE's ability to adaptively incorporate judges for improved supervision.

### 5.4. Robustness to Confounding Factors

**Setup.** We assess CARE's robustness using the dataset from Chen et al. (2024), where LLM responses are systematically altered through targeted GPT-4 prompts. We focus on two stylistic biases that preserve semantic correctness: *beauty*, which adds superficial emojis or formatting, and *authority*, which includes a fake citation. LLM judges assign scores from 1 to 10 for each response. Robustness is measured by comparing aggregated scores before and after perturbation using MAE, where lower values indicating greater robustness.

**Results.** Table 3 shows that CARE achieves strong robustness against beauty and authority biases, with the lowest MAE compared to baseline methods. This indicates that CARE effectively mitigates superficial stylistic changes, such as cosmetic formatting or fake citations, while maintaining consistent aggregated evaluations.

### 5.5. Defending Against Adversarial Responses

LLM judges have been shown to be vulnerable to simple token manipulations (e.g., ":" or " ") and reasoning prompts (e.g., "Let's think step by step"), often interpreting them as signs of correctness (Zhao et al., 2025). We hypothesize that CARE can mitigate these vulnerabilities by serving as a defense mechanism against such adversarial triggers.

**Setup.** We evaluate CARE on adversarial responses from Zhao et al. (2025) using the multi-subject RLVR dataset (Su et al., 2025). Each judge is presented with an adversarial response paired with the true reference answer and asked to assess its correctness. Since adversarial answers are inher-

ently incorrect, any judgment labeling them as correct is a false positive. We compare MV, WS, and CARE, reporting the false positive rate (lower is better).

**Results.** Table 4 reports false positive rates across different adversarial response types. CARE consistently reduces the false positive rate, often by a substantial margin, without requiring additional interventions.

## 6. Related Work

We review prior work on *bias in LLM-as-a-judge* and *label aggregation*, and position our method in relation to both. An extended discussion appears in Appendix B.

**Bias in LLM-as-a-judge.** LLMs used as automated evaluators exhibit systematic preferences, including positional, verbosity, authority, and self-enhancement biases (Ye et al., 2025; Zhu et al., 2025). Prior work mitigates these effects primarily at the level of *individual judges*, through prompt-based interventions (Shi et al., 2024; Jiao et al., 2024; Ye et al., 2025) or fine-tuned evaluators such as JudgeLM and PandaLM that better align judgments with human preferences (Zhu et al., 2025; Wang et al., 2024b; Li et al., 2024d). In contrast, we target the downstream multi-judge setting: *rather than debiasing individual LLMs, we explicitly model shared latent confounders that induce correlated errors across judges, enabling principled aggregation even when single-judge biases persist.*

**Label Aggregation.** Classic aggregation models such as Dawid–Skene (Dawid & Skene, 1979), GLAD (Whitehill et al., 2009), and MACE (Hovy et al., 2013) infer latent truth by modeling annotator-specific noise. Weak supervision frameworks extend this paradigm to programmatic sources (Bach et al., 2019a; Fu et al., 2020; Shin et al., 2022). More recently, (Hu et al., 2024b) propose GED, which ensembles and denoises preference graphs from multiple weak LLM evaluators, while Wang et al. (2025) analyze inference and aggregation statistics for LLM-as-a-judge outputs. These methods typically assume conditional independence or unstructured noise; our approach departs from this assumption by *explicitly captures shared latent factors (e.g., verbosity or formality) affecting all judges, bridging single-judge bias mitigation and statistical aggregation.*

# 7. Conclusion

We introduce CARE, a confounder-aware aggregation framework that formulates multi-judge scoring in a higher-rank latent-variable model and delivers three main contributions. **(i)** It explicitly models shared confounders, providing an aggregation scheme for LLM-judge scenarios. **(ii)** It offers statistically principled estimators—sparse-plus-low-rank covariance recovery and tensor method—with provable identifiability. **(iii)** It demonstrates consistent empirical gains, improving accuracy and robustness across diverse public benchmarks.

# Acknowledgements

We are grateful for the support of the National Science Foundation (NSF) (CCF2106707), the Defense Advanced Research Projects Agency (DARPA Young Faculty Award), and the Wisconsin Alumni Research Foundation (WARF).

# Impact Statement

This work presents a novel approach to aggregate scores from multiple LLMs serving as judges by identifying confounding variables and thus potentially reducing the bias in the overall judge scores. The potential broader impact includes a framework for improved LLM-as-a-judge scores which can be used at various applications. However, it is important to acknowledge that using LLMs as potential judges to automate labor-intense annotation tasks which is an active area of research carries some limitations and past research has discussed some unintended consequences, such as over-reliance on judge outputs, misuse and misinterpretation of results which might carry high real-world stakes. It is crucial to use automated LLM-as-a-judge tools responsibly and ethically, considering potential biases in data and models, and ensuring transparency and accountability in their application.

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

The appendix is structured as follows. It starts with the glossary table, defining key notations used throughout the paper in Appendix A. Next, Appendix B discusses additional related work. In Appendix C, we introduce details about our tensor-based CARE algorithm, discussion for general CARE method, and additional discussion about method heuristics. Following this, Appendix D offers theoretical support of our approach and supported proofs. It includes the graphical model formulation, graph structure recovery error bound, sample complexity, and the misspecification error arising from incorrectly characterized confounding factors. Subsequently, Appendix E provides experimental details and additional experiment results. Finally, Appendix 7 concludes by discussing the broader impacts and limitations of the work.

# A. Glossary

The notations are summarized in Table 5.

*Table 5.* Glossary of variables and symbols used in this paper.

| Symbol | Definition |
|---|---|
| $(J_1, \ldots, J_p)$ | $p$ vector of Judges score |
| $Q$ | True-quality latent variable |
| $(C_1, \ldots, C_k)$ | $k$ latent confounder variables |
| $H$ | All the hidden variables (true + confounder) i.e $(Q\ C_1, \ldots, C_k)$ |
| $h$ | dimension of $H$ i.e all hidden variables $= k + 1$ |
| $X$ | Score matrix of dimension $(n \times p)$ where $n$ is the number of examples and $p$ is the number of judges |
| $K$ | Precision matrix |
| $K_{oo}$ | Observable-observable connection matrix |
| $K_{oh}$ | Observable-latent connection matrix |
| $K_{hh}$ | Latent-latent connection matrix |
| $\Sigma_o$ | Covariance matrix of observable variables |
| $S$ | Sparse matrix ($\mathbb{R}^{p \times p}$) which encodes edges between judges |
| $L$ | Low-rank matrix (with $rank(L) \le h$) which captures dependencies mediated by latent variables |
| $R$ | Rotation matrix ($\mathbb{R}^{h \times h}$) |
| $\gamma_n$ | Regularization for sparse and low-rank matrix $S$ in Algorithm 1 |
| $\tau$ | Regularization for low-rank matrix $L$ in Algorithm 1 |
| $\hat{s}_{\text{agg}}^{(i)}$ | Aggregated scores for $i$th example in the dataset from $p$ judges |
| $\hat{\Sigma}$ | Sample covariance matrix |
| $\hat{\Theta}$ | Sample precision matrix |
| $\hat{S}$ | Sample Sparse matrix ($\mathbb{R}^{p \times p}$) which encodes direct connectional edges among judges |
| $\hat{L}$ | Sample Low-rank matrix (with $rank(L) \le h$) which captures dependencies mediated by latent variables |
| $U$ | Latent factor extraction matrix i.e latent-judge connections ($\mathbb{R}^{p \times h}$) from Algorithm 1 |
| $\Theta$ | Precision matrix |
| $w$ | Weight for aggregating judges |
| $\lambda$ | Singular values of $L$ |
| $u^\star$ | Singular vector of $L$ corresponds to true quality factor |
| $\lambda^\star$ | Singular value of $L$ that corresponds to true quality factor |
| $\mu_{qc}$ | Conditional mean of judges given $Q = q, C = c$ |
| $\pi_{qc}$ | Probability of $Q = q, C = c$ |
| $\alpha_{qc}(J)$ | Posterior responsibility $\Pr(Q{=}q, C{=}c \mid J)$. Used to aggregate component-wise inferences |
| $\phi(J; \mu, \Sigma)$ | Gaussian density used in Bayes updates |
| $\hat{\mu}_{qc}$ | Estimated conditional mean of judges given $Q = q, C = c$ |
| $\hat{\pi}_{qc}$ | Estimation of probability of $Q = q, C = c$ |
| $\{\mathcal{G}_\ell\}_{\ell=1}^3$ | Groups of judges that are independent of judges outside the group |
| $\hat{T}$ | Empirical 3-way tensor |
| $\hat{\mu}_{qc}^{(1)}, \hat{\mu}_{qc}^{(2)}, \hat{\mu}_{qc}^{(3)}$ | Estimated conditional mean of three views |
| $\hat{\mu}_{\rho(r)}$ | Estimated conditional mean of judges after permutation |
| $\mu_{\text{anchor}(r)}$ | Conditional mean of anchor sets |

# B. Extended Related Work

Here we discuss other related works, including biases in the use of LLM-as-a-judge, and aggregation methods in the weak supervision literature.

## B.1. Biases in LLM–as–a–Judge

LLMs have rapidly become the standard tools for automatically evaluating generation tasks, as their judgments show strong correlations with those of human annotators—particularly in areas such as translation and summarization (Kocmi & Federmann, 2023; Shen et al., 2023; Chiang & yi Lee, 2023). Yet a growing body of work shows that these models are far from impartial. **Positional bias**—preferring the *second* answer in a pairwise comparison—was first noted in MT-Bench (Zheng et al., 2023a) and later quantified in detail by (Wang et al., 2024a), who observed reversals of up to 30% when simply swapping order. **Verbosity bias**, wherein longer answers receive higher scores regardless of quality, is highlighted by (Chen et al., 2024). LLM judges also display **self-enhancement bias**, overrating responses produced by models from the same family (Zeng et al., 2024). Less studied but equally problematic are **concreteness/authority biases**: judges over-reward answers that contain citations, numbers, or confident tone even when these features are irrelevant (Park et al., 2024).

Mitigation strategies span two levels. *Prompt-level interventions* randomize answer order, enforce symmetric formatting, and instruct the judge to ignore superficial features (Wang et al., 2024a; Li et al., 2024d). Adding chain-of-thought rationales or decomposing the rubric into sub-criteria (accuracy, conciseness, style) also moderates shallow heuristics (Khan et al., 2024). On the *model level*, fine-tuned evaluators such as JudgeLM (Zhu et al., 2025) and Split-and-Merge Judge (Li et al., 2024d) are trained on curated data that explicitly counter positional and length biases. Peer-review and debate schemes go a step further: PRD lets a second LLM critique the first judge and often corrects biased decisions (Li et al., 2024c), while (Khan et al., 2024) show that dialog with a more persuasive model leads to more truthful verdicts.

Despite progress, most debiasing work treats a *single* judge in isolation. When evaluations aggregate many LLM scorers—for robustness, cost sharing, or diversity—*biases can compound in complex ways that individual fixes do not capture*.

## B.2. Label Aggregation for Multiple Noisy Evaluators

**Weak-supervision.**    Treating each LLM prompt or model as a noisy *labeling function* aligns aggregation with modern weak supervision. Snorkel (Ratner et al., 2017; Bach et al., 2019b) estimates source accuracies and dependencies to denoise programmatic labels, laying the foundation for LLM-prompt aggregation. Across domains, weak supervision has powered industrial-scale text classification and content understanding, knowledge-base construction and distant-supervision relation extraction, multi-resolution labeling of sequential video/sensor streams, and weakly supervised medical-image segmentation (Ratner et al., 2017; Bach et al., 2019b; Niu et al., 2012; Mintz et al., 2009; Varma et al., 2019; Hooper et al., 2021). On more technical side, Fu et al. (2020) introduces a scalable moment-matching estimator with closed-form weights. (Shin et al., 2022) generalizes label models beyond categorical labels to arbitrary metric spaces, greatly expanding their applicability. Cachay et al. (2021) jointly optimizes a classifier and a differentiable label model, outperforming two-stage pipelines when sources are dependent. Firebolt further removes requirements on known class priors or source independence, estimating class-specific accuracies and correlations in closed form (Kuang et al., 2022). Shin et al. (2023) shows that fixing source bias in labeling functions using optimal transport can improve both accuracy and fairness.

**Aggregation of multiple *LLM* judges.**    Recent work shows that *ensembling smaller evaluators can beat a single large judge*. The **PoLL** jury combines three diverse 7–35B models and attains higher correlation with human ratings than GPT-4 while costing 7× less and reducing bias (Verga et al., 2024b). **GED** merges preference graphs from weak evaluators (Llama3-8B, Mistral-7B, Qwen2-7B) and denoises cycles; its DAG ranking surpasses a single 72B judge on ten benchmarks (Hu et al., 2024a). **JudgeBlender** ensembles either multiple models or multiple prompts, improving precision and consistency of relevance judgments over any individual LLM (Rahmani et al., 2024). These findings echo classic "wisdom-of-crowds" results—when paired with principled aggregation, a panel of smaller, heterogeneous judges can outperform a much larger model, offering a practical path toward reliable, low-cost evaluation.

## B.3. Our Contribution in Context

Prior research either (i) debiases one judge at a time or (ii) aggregates multiple judges assuming independent noise. Our confounder-aware aggregation unifies these threads. We posit latent factors (e.g., verbosity, formality) that influence *all*

judges simultaneously and show how to infer both the latent truth and the shared confounders. This yields more reliable consensus scores when individual judges—human or LLM—share systemic biases.

## C. Algorithm Details

This section details the implementation of our CARE framework. Specifically, it includes the full CARE tensor algorithm, details about SVD baseline method for comparing our tensor-based algorithm, generalizations beyond Gaussian assumptions, and practical heuristics to address non-orthogonality in latent factors and justification for where the sparse structure lies in mixed Gaussian data.

### C.1. Tensor-based CARE Algorithm

### C.2. SVD Baseline in Synthetic Experiment

We form the empirical two-way moment between view 1 and view 2:

$$\widehat{M}_{1,2} = \frac{1}{n} \sum_{i=1}^{n} X_1^{(i)} X_2^{(i)\top}$$
$$= \sum_{q,c} \pi_{q,c}\, \mu_{1,q,c}\, \mu_{2,q,c}^{\top} \ + \ \text{sampling noise},$$

where $\pi_{q,c} = \Pr[Q = q, C = c]$ and $\mu_{v,q,c} = E[J_v \mid Q = q, C = c]$ for judge/view $v$ A singular-value decomposition

$$\widehat{M}_{1,2} \ = \ U_{12}\, \Sigma_{12}\, V_{12}^{\top}$$

yields factor matrices

$$U_{12}\, \Sigma_{12}^{1/2} \approx [\mu_{1,q,c}]\, R, \quad V_{12}\, \Sigma_{12}^{1/2} \approx [\mu_{2,q,c}]\, R,$$

where $R \in O(4)$ is an unknown orthogonal matrix.

Repeating on $\widehat{M}_{1,3} = \frac{1}{n} \sum_i X_1^{(i)} X_3^{(i)\top} = U_{13}\, \Sigma_{13}\, V_{13}^{\top}$ produces a second rotated copy of $[\mu_{1,q,c}]$. We solve the Procrustes problem

$$R \ = \ \arg \min_{O \in O(4)} \big\| U_{12}\, \Sigma_{12}^{1/2} \ - \ U_{13}\, \Sigma_{13}^{1/2}\, O \big\| * F,$$

then set $\hat{\mu}_{2,q,c} = (V_{12}\, \Sigma_{12}^{1/2})\, R^{\top}$ and $\hat{\mu}_{3,q,c} = (V_{13}\, \Sigma_{13}^{1/2})\, R^{\top}$ to align all three views.

This SVD baseline recovers $\{\mu_{v,q,c}\}$ up to the permutation/sign ambiguity inherent in any orthogonal transform.

### C.3. General CARE Setup

**Extension Beyond the Gaussian Observation Model.** The multivariate-Gaussian assumption for $J|H$ is convenient—its first two or three moments already encode all information needed for the sparse + low-rank and tensor steps—but it is not a requirement. Because CARE learns the *graphical* structure, the same pipeline applies whenever each judge's conditional distribution lies in an exponential family or, more generally, a latent-variable generalized linear model (GLM):

- **Categorical or ordinal scores.** For Likert ratings or pairwise preferences we can set

$$J_i \mid H \ \sim \ \begin{cases} \text{Bernoulli}\big(\sigma(W_i^{\top} H)\big), & \text{(Binary / pairwise preference)}, \\ \text{Categorical}\big(\text{softmax}(W_i^{\top} H)\big), & \text{(Categorical)}, \\ \text{OrderedLogit}\big(W_i^{\top} H; \boldsymbol{\tau}\big), & \text{(Ordinal / Likert)}. \end{cases} \tag{2}$$

Here $\sigma(t) = 1/(1 + e^{-t})$. For $\text{OrderedLogit}(\eta; \boldsymbol{\tau})$ with $\eta = W_i^{\top} H$ and $-\infty = \tau_0 < \tau_1 < \cdots < \tau_{R-1} < \tau_R = +\infty$, we use

$$\Pr(J_i \leq r \mid H) = \sigma(\tau_r - \eta), \qquad \Pr(J_i = r \mid H) = \sigma(\tau_r - \eta) - \sigma(\tau_{r-1} - \eta).$$

---

**Algorithm 2** CARE-Tensor

---

**Require:** Score matrix $J \in \mathbb{R}^{n \times p}$, tolerance $\varepsilon$.
**Ensure:** Estimates $\{\hat{\mu}_{qc}, \hat{\pi}_{qc}\}_{q,c \in \{0,1\}}$.

1: **A. Anchor discovery (graph partition)**
2: Compute the sample covariance $\hat{\Sigma} = J^{\top} J / n$
3: Compute the precision $\hat{\Theta} = \hat{\Sigma}^{-1}$
4: Perform the sparse+low-rank decomposition $\hat{\Theta} \approx \hat{S} - \hat{L}$ (Alg. 1 Step 2).
5: Partition judges into three disjoint groups $\{\mathcal{G}_\ell\}_{\ell=1}^3$ that satisfy

$$a \neq b, \; j_1 \in \mathcal{G}_a, \; j_2 \in \mathcal{G}_b \implies |\hat{S}_{j_1, j_2}| \leq \varepsilon,$$

   ensuring no direct edges with strength greater than $\varepsilon$ can exist across groups.

6: **B. Empirical third-order moment tensor**
7: **for** $\ell = 1, 2, 3$ **do**
8:     $X_\ell \leftarrow$ columns of $J$ indexed by $\mathcal{G}_\ell$ $\{X_\ell \in \mathbb{R}^{n \times |\mathcal{G}_\ell|}\}$
9: **end for**
10: Compute

$$\hat{T} = \frac{1}{n} \sum_{i=1}^{n} X_1^{(i)} \otimes X_2^{(i)} \otimes X_3^{(i)} \; \in \; \mathbb{R}^{|\mathcal{G}_1| \times |\mathcal{G}_2| \times |\mathcal{G}_3|}.$$

11: **C. Tensor decomposition**
12: Run a CP tensor-power decomposition on $\hat{T}$ to obtain $k = 4$ components
   $\left\{(\hat{\pi}_{qc}, \hat{\mu}_{qc}^{(1)}, \hat{\mu}_{qc}^{(2)}, \hat{\mu}_{qc}^{(3)})\right\}_{q,c \in \{0,1\}^2}$, where $\hat{\pi}_{qc} > 0$ and $\hat{\mu}_{qc}^{(\ell)} \in \mathbb{R}^{|\mathcal{G}_\ell|}$.
13: **D. Assemble full means**
14: **for** $q, c \in \{0,1\}^2$ **do**
15:     $\hat{\mu}_{qc} \leftarrow \text{concat}(\hat{\mu}_{qc}^{(1)}, \hat{\mu}_{qc}^{(2)}, \hat{\mu}_{qc}^{(3)}) \in \mathbb{R}^p$.
16: **end for**
17: **E. Unsupervised state identification (quality axis)**
18: $v \leftarrow \text{TOPEIGENVECTOR}(\hat{L})$.
19: $s_r \leftarrow v^{\top} \hat{\mu}_r \quad (r = 1, \ldots, 4)$.
20: $\mathcal{S}_{Q=1} \leftarrow \text{TOPTWOINDICES}(s_1, \ldots, s_4)$; $\mathcal{S}_{Q=0} \leftarrow \{1, 2, 3, 4\} \setminus \mathcal{S}_{Q=1}$. {two highest scores $\rightarrow Q{=}1$; remainder $\rightarrow Q{=}0$}
21: **F. Optional state alignment with anchors**
22: Find the permutation $\rho$ of $\{1, \ldots, 4\}$ that minimizes $\sum_{r=1}^4 \|\hat{\mu}_{\rho(r)} - \mu_{\text{anchor}(r)}\|_2^2$, where the four anchor prototypes correspond to $(Q, C) = \{00, 01, 10, 11\}$.
23: $(\hat{\mu}_{00}, \hat{\mu}_{01}, \hat{\mu}_{10}, \hat{\mu}_{11}) \leftarrow (\hat{\mu}_{\rho(1)}, \hat{\mu}_{\rho(2)}, \hat{\mu}_{\rho(3)}, \hat{\mu}_{\rho(4)})$.
24: **G. Mixing weights**
25: $(\hat{\pi}_{00}, \hat{\pi}_{01}, \hat{\pi}_{10}, \hat{\pi}_{11}) \leftarrow (\hat{\pi}_{\rho(1)}, \hat{\pi}_{\rho(2)}, \hat{\pi}_{\rho(3)}, \hat{\pi}_{\rho(4)})$.
26: **return** $\{\hat{\mu}_{qc}, \hat{\pi}_{qc}\}_{q,c \in \{0,1\}}$.

---

The graph—hence the sparse mask $S$—is unchanged; only the node-wise likelihoods differ. We still recover $S$ from conditional-mutual-information or pseudo-likelihood scores, and we still factorize higher-order indicator moments such as $\mathbb{E}\left[\mathbf{1}_{\{J_a=\ell\}}\,\mathbf{1}_{\{J_b=m\}}\,\mathbf{1}_{\{J_c=n\}}\right]$.

- **Mixed Discrete-Continous Scores.** When some judges output real scores and others categorical flags, we use a mixed conditional distribution:

$$p(J \mid H) = \prod_{i \in \text{Cont.}} \mathcal{N}(J_i; \mu_{H_i}, \sigma_i^2) \prod_{j \in \text{Disc.}} \text{Bernoulli}(\sigma(W_j^\top H)).$$

CARE forms mixed raw/indicator moments, and identifiability again follows from standard tensor-decomposition guarantees for mixed conditional means.

- **Heavy-tailed or skewed real scores.** When numeric scores are skewed or contain outliers, a multivariate-$t$ or Gaussian scale mixture is appropriate. Up to a scalar factor, the covariance still decomposes as sparse + low-rank, so Steps 1–2 of Algorithm 1 work after a simple rescaling.

Empirically, we find that replacing the Gaussian local likelihood only affects the estimation of sparse structure and extraction of latent factors, not the subsequent symmetry-breaking or posterior computation; thus the overall CARE pipeline generalizes with minimal adjustments.

### C.4. Heuristics and Justifications

**More Heuristics for Symmetry Breaking in CARE** For CARE-SVD, we introduce two complementary heuristics to distinguish the true-quality latent factor from confounders. First, the *human-anchor criterion* uses a small validation set with human ratings: among the estimated factors, we select the one whose loadings best agree with human judgments. Second, a *loading-balance heuristic* prefers a factor that loads broadly and with comparable magnitude across competent judges, while rejecting factors dominated by a small subset (often judge-specific confounders, e.g., length-sensitive factor).

Additionally for CARE-Tensor, if high-quality labeled samples are available (e.g. known samples with $Q = 1$), we can anchor the configuration by aligning the state whose conditional means best match labeled $Q{=}1$ samples. Without labels, we automate this by taking the leading eigenvector $v$ of the learned low-rank factor $\hat{L}$ (a proxy for the shared "quality" axis), scoring each recovered conditional mean $\hat{\mu}_r$ via $s_r = v^\top \hat{\mu}_r$, and assigning the highest-scoring state(s) to $Q{=}1$.

**Heuristic for Addressing Orthogonality Violations in CARE (SVD).**

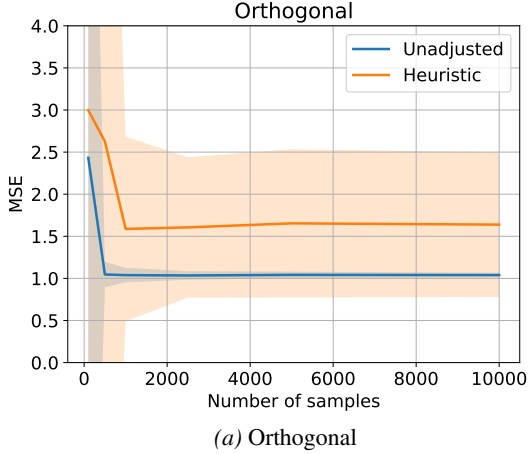
*(a)* Orthogonal

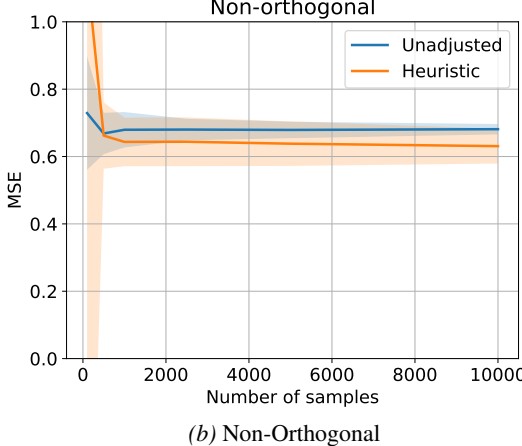
*(b)* Non-Orthogonal

*Figure 5.* Effect of the proposed heuristic in a fully Gaussian synthetic setup. We estimate the true quality variable $Q$ and report the mean squared error. The heuristic improves estimation in the non-orthogonal setting, but slightly degrades performance in the orthogonal setting where true and confounding components are disjoint.

Existing heuristics for identifying the true quality latent factor can estimate corresponding weights, but they often suffer from bias when the orthogonality assumption—central to the application of SVD—is violated. This issue commonly arises

in real-world datasets. We found the following weighting rule effective in both synthetic and real-world settings:

$$w \leftarrow \lambda^\star u^\star - \sum_{u_i \in U \setminus \{u^\star\}} \lambda_i u_i,$$

where $w$ represents the learned weights for each judge, $\lambda^*$ and $u^*$ is the singular value and vector of $L$ that corresponds to the direction that is closest to true quality latent variable, $\lambda_i, u_i$ represent rest of the singular values and vectors, which can be interpreted as spurious/confounding factors.

This rule intuitively subtracts the influence of overlapping (non-orthogonal) confounding components from the estimated true score factor.

Figure 5 illustrates the effect of this heuristic in a synthetic fully Gaussian setup. In the non-orthogonal case—where confounding components overlap with the true signal—the heuristic improves the estimation of the true latent variable. In contrast, it underperforms in the orthogonal case, where judges influenced by true scores are cleanly separated from those influenced by confounders.

# D. Theory

We formalize the graphical model under joint gaussian distribution and notation (Section D.1), then discuss the identifiability of graph structure with exact and approximate recovery (Section D.2) and quantify the sample complexity required for consistent recovery of our SVD-based algorithm (Section D.3). Next, we present the model misspecification error when confounding factor is not correctly characterized (Section D.4). Finally, we discuss sample complexity required for tensor-based algorithm under mixed Gaussian distribution (Section D.5. All proofs are included in Section D.6.

## D.1. Model and Notation

We discuss the model under joint-gaussian distribution where all variables follow the same definitions as in Section 3. Briefly, $J = (J_1, \ldots, J_p)^\top$ stacks the $p$ observable judge scores, and $H = (Q, C_1, \ldots, C_k)^\top$ collects the $h = k + 1$ latent variables.

$$\Sigma = \mathrm{Cov}[(J, H)^\top], \qquad \Sigma^{-1} = K = \begin{pmatrix} K_{JJ} & K_{JH} \\ K_{HJ} & K_{HH} \end{pmatrix},$$

where the subscript $J$ (resp. $H$) refers to observable (resp. latent) coordinates.

The observable block factorizes via the Schur complement:

$$(\Sigma_{JJ})^{-1} = S - L, \quad S = K_{JJ}, \quad L = K_{JH} K_{HH}^{-1} K_{HJ}.$$

Here $\Sigma_o$ is the covariance matrix of observable variables, $S \in \mathbb{R}^{p \times p}$ is sparse and encodes direct conditional edges among judges, $L$ is low-rank with $rank(L) \le h$ and captures dependencies mediated by the latent variables. Entry $(K_{JH})_{i\ell}$ is the edge weight between judge $i$ and latent factor $\ell$.

## D.2. Graph Structure Identifiability

While $(S, L)$ can be recovered (e.g. via convex sparse-plus-low-rank regularization (Chandrasekaran et al., 2012), the finer structure of $K_{JH}$ is usually not identifiable from $L$. For example, for arbitrary rotation matrix $R \in \mathbb{R}^{h \times h}$, $L = (K_{JH} K_{HH}^{-1/2} R)(R^\top K_{HH}^{-1/2} K_{HJ})$, this indicates one cannot distinguish $K_{JH} K_{HH}^{-1/2}$ from $K_{JH} K_{HH}^{-1/2} R$ without further constraints. Hence, we need to impose additional assumptions:

**Assumption D.1** (Latent–latent independence and eigengap in $K_{HH}^{-1}$). $K_{HH}^{-1} = \mathrm{diag}(\lambda_1, \ldots, \lambda_h)$ with $\lambda_1 > \lambda_2 > \cdots > \lambda_h > 0$. (Equivalently, $K_{HH} = \mathrm{diag}(1/\lambda_1, \ldots, 1/\lambda_h)$.)

**Assumption D.2** (Orthonormal latent–observable connections). The columns of $K_{JH}$ are orthonormal, i.e. $K_{JH}^\top K_{JH} = I_h$. A special case is the *disjoint-support* model where each judge connects to exactly one latent factor.

Next, we provide an exact recovery result given the above assumptions.

**Theorem D.3** (Exact Recovery). *Under Assumptions 1 and 2, columns in $K_{JH}$ are identifiable up to column permutations and sign flips.*

Real-world data rarely satisfy the exact orthogonality in Assumption D.2. To assess robustness, consider the following perturbed connection matrix:

$$\tilde{K}_{JH} = K_{JH} + E, \qquad \|E\|_2 \text{ small}.$$

The associated low-rank part is $\tilde{L} = \tilde{K}_{JH} K_{HH}^{-1} \tilde{K}_{HJ}$. Let the eigen-pairs of $L = K_{JH} K_{HH}^{-1} K_{HJ}$ and $\tilde{L}$ be $\{(\lambda_i, u_i)\}_{i=1}^h$ and $\{(\tilde{\lambda}_i, \tilde{u}_i)\}_{i=1}^h$, ordered so that $\lambda_1 > \cdots > \lambda_h > 0$, and denote the eigen-gap by

$$\delta_i = \min_{j \neq i} |\lambda_i - \lambda_j| > 0.$$

**Theorem D.4** (Stability under approximate orthogonality). *Assume Assumptions D.1–D.2 and that $K_{JH}$ has orthonormal columns. Let $\tilde{K}_{JH} = K_{JH} + E$ with $\|E\|_2$ small, and define $\tilde{L} = \tilde{K}_{JH} K_{HH}^{-1} \tilde{K}_{HJ}$. Let $\{(\lambda_i, u_i)\}_{i=1}^h$ and $\{(\tilde{\lambda}_i, \tilde{u}_i)\}_{i=1}^h$ be the top-$h$ eigenpairs of $L$ and $\tilde{L}$, ordered so $\lambda_1 > \cdots > \lambda_h > 0$. Define the (full-spectrum) eigengap*

$$\delta_i := \min\Big\{\lambda_i, \min_{j \in [h],\, j \neq i} |\lambda_i - \lambda_j|\Big\}.$$

*Then for each $i \in [h]$, there exists a sign $s_i \in \{\pm 1\}$ such that*

$$\|\tilde{u}_i - s_i u_i\|_2 \;\leq\; \frac{4 \, \|K_{HH}^{-1}\|_2 \, \|E\|_2}{\delta_i} \;+\; O\big(\|E\|_2^2\big).$$

### D.3. Sample Complexity Bound

We now quantify how many i.i.d. samples are needed for the two–stage estimator in Algorithm 1 to recover the latent–observable directions $K_{JH} \in \mathbb{R}^{p \times h}$.

As detailed in Algorithm 1, our estimator for $K_{JH}$ proceeds in two stages: first, a sparse + low-rank decomposition of sample precision matrix. Second, we extract the latent–observable directions by taking the rank-$h$ eigen-decomposition $\hat{L}_n = \sum_{i=1}^h \hat{\lambda}_i \hat{u}_i \hat{u}_i^\top$ and setting $\hat{K}_{JH} := [\hat{u}_1, \ldots, \hat{u}_h]$.

**Theorem D.5** (Sample complexity for recovering $K_{JH}$). *Let $L^* = K_{JH} K_{HH}^{-1} K_{HJ} \in \mathbb{R}^{p \times p}$ have distinct eigenvalues $\lambda_1 > \cdots > \lambda_h$ and define the (global) eigengap $\delta := \min_{1 \leq i < j \leq h} |\lambda_i - \lambda_j|$. Assume the identifiability, incoherence, and curvature conditions of (Chandrasekaran et al., 2012). Then for any $\eta > 0$, with probability at least $1 - 2e^{-\eta}$,*

$$\max_{i \leq h} \big\| \hat{u}_i - u_i \big\|_2 = O\Big(\frac{\sqrt{\eta}}{\sqrt{n} \, \xi(T) \, \delta}\Big),$$

*where $n$ is the sample size and $T = T(L^*)$.*

We defer the proof to Appendix D.6. At a high-level, we adapt the identifiability, incoherence and curvature conditions from Theorem 4.1 of (Chandrasekaran et al., 2012) and combine it with extended result of Davis-Khan's theorem (Yu et al., 2015).

This bound shows that the column-wise $\ell_2$ error decays at the standard parametric rate $n^{-1/2}$, and is attenuated by both the manifold curvature $\xi(T)$ and the eigengap $\delta$. Achieving an accuracy of at most $\alpha \in (0, 1)$ therefore requires

$$n = \tilde{O}\Big(\frac{\epsilon}{\xi(T)^2 \delta^2 \alpha^2}\Big)$$

samples, up to universal constants and log-factors.

### D.4. Misspecification Error

Many label aggregation frameworks (e.g.,(Bach et al., 2019a; Fu et al., 2020; Shin et al., 2022)) assume a *single* latent variable that explains the observed labels. However, in setups like LLM-as-a-judge, the scores may be influenced by additional latent factors or confounders that also affect the observed annotations. Ignoring these *confounder* latents leads to model misspecification, which can bias the aggregated labels. We characterize this bias and analyze its impact on the estimated aggregation weights.

Let $L^* = \sum_{\ell=1}^h \frac{1}{d_\ell} \mathbf{k}_\ell \mathbf{k}_\ell^T$ be the true rank-$h$ low-rank component of the observable precision matrix, derived from the latent-observable connection matrix $K_{JH} = [\mathbf{k}_1, \ldots, \mathbf{k}_h]$ and latent-latent precision $K_{HH} = \text{diag}(d_1, \ldots, d_h)$. Let $\mathbf{u}_1^{\text{true}} = \mathbf{k}_1 / \|\mathbf{k}_1\|_2$ be the true direction of influence for the quality score latent variable $Q$ (assuming $\mathbf{k}_1 \neq \mathbf{0}$).

Define $\mathbf{A} = \frac{1}{d_1}\mathbf{k}_1\mathbf{k}_1^T$. Its principal (and only non-zero) eigenvalue is $\lambda_1 = \frac{1}{d_1}||\mathbf{k}_1||_2^2$, and its spectral gap (to its other zero eigenvalues) is $\delta = \lambda_1$. Let $\mathbf{E} = \sum_{\ell=2}^h \frac{1}{d_\ell}\mathbf{k}_\ell\mathbf{k}_\ell^T$ be the confounding component, so $L^* = \mathbf{A} + \mathbf{E}$. Let $\mathbf{v}_1$ be the principal unit-norm eigenvector of $L^*$. When a rank-1 model is fitted, the estimated direction is $\hat{\mathbf{u}}_1^{\text{pop}} = \mathbf{v}_1$.

**Theorem D.6.** *If $||\mathbf{E}||_{op} \leq \delta/2$, the $\ell_2$ deviation of the estimated direction $\mathbf{v}_1$ from $\mathbf{u}_1^{\text{true}}$ is bounded by:*

$$\left|\left|\mathbf{v}_1 - s\mathbf{u}_1^{\text{true}}\right|\right|_2 \leq \frac{2\,||\mathbf{E}||_{op}}{\delta} = \frac{2\left|\left|\sum_{\ell=2}^h \frac{1}{d_\ell}\mathbf{k}_\ell\mathbf{k}_\ell^T\right|\right|_{op}}{\frac{1}{d_1}||\mathbf{k}_1||_2^2}$$

*for a sign $s = \pm 1$ (chosen so that $s(\mathbf{u}_1^{\text{true}})^T\mathbf{v}_1 \geq 0$).*

*Proof.* By Davis-Kahan theorem (Theorem 2 in (Yu et al., 2015)), if $||\mathbf{E}||_{op} \leq \delta/2$, then the $\ell_2$ distance between $\mathbf{v}_1$ and $\mathbf{u}_1^{\text{true}}$ (after aligning their signs via $s = \pm 1$) is bounded by:

$$\left|\left|\mathbf{v}_1 - s \cdot \mathbf{u}_1^{\text{true}}\right|\right|_2 \leq \frac{2\,||\mathbf{E}||_{\text{op}}}{\delta}.$$

Plugging in $E$ yields the desired result:

$$\left|\left|\mathbf{v}_1 - s \cdot \mathbf{u}_1^{\text{true}}\right|\right|_2 \leq \frac{2\left|\left|\sum_{\ell=2}^h \frac{1}{d_\ell}\mathbf{k}_\ell\mathbf{k}_\ell^T\right|\right|_{\text{op}}}{\frac{1}{d_1}||\mathbf{k}_1||_2^2}.$$

$\square$

The theorem quantifies the directional bias in the estimated influence of $Q$ when confounders are ignored. This bias is proportional to the collective "strength" of confounders in the precision domain (numerator) and inversely proportional to $Q$'s own "strength" (denominator). Fitting a rank-1 model forces this bias, while a higher-rank model offers the capacity to separate these influences.

**Corollary D.7** (Error Bound for Estimated Conditional Mean of $Q$). *Denote the true conditional mean of true quality score latent variable $Q$ given the observable variables $O = (J_1, ..., J_p)$ be denoted by $\mathbb{E}[Q|O]_{\text{true}}$. Then, $\mathbb{E}[Q|\mathbf{o}]_{\text{true}} = -\frac{||\mathbf{k}_1||_2}{d_1}(\mathbf{u}_1^{\text{true}})^T\mathbf{o}$. Let an estimated conditional mean with the misspecified direction, $\mathbb{E}[Q|\mathbf{o}]_{\text{mis}}$, be formed using the misspecified direction $\mathbf{v}_1$ be $\mathbb{E}[Q|\mathbf{o}]_{\text{mis}} = -\frac{||\mathbf{k}_1||_2}{d_1}(s \cdot \mathbf{v}_1)^T\mathbf{o}$, where $s = \pm 1$ is chosen such that $s \cdot (\mathbf{u}_1^{\text{true}})^T\mathbf{v}_1 \geq 0$. Then, the absolute error in the estimated conditional mean due to the directional misspecification is bounded by:*

$$|\mathbb{E}[Q|\mathbf{o}]_{\text{mis}} - \mathbb{E}[Q|\mathbf{o}]_{\text{true}}| \leq \frac{2\left|\left|\sum_{\ell=2}^h \frac{1}{d_\ell}\mathbf{k}_\ell\mathbf{k}_\ell^T\right|\right|_{op}}{||\mathbf{k}_1||_2}||\mathbf{o}||_2$$

*This holds if the condition from the main theorem, $||\mathbf{E}||_{op} \leq \delta/2 = \frac{1}{2d_1}||\mathbf{k}_1||_2^2$, is met, where $\mathbf{E} = \sum_{\ell=2}^h \frac{1}{d_\ell}\mathbf{k}_\ell\mathbf{k}_\ell^T$.*

*Proof.* The absolute difference is:

$$\left|\mathbb{E}[Q\,|\,\mathbf{o}]_{\text{mis}} - \mathbb{E}[Q\,|\,\mathbf{o}]_{\text{true}}\right| = \frac{||\mathbf{k}_1||_2}{d_1}\left|(s\mathbf{v}_1 - \mathbf{u}_1^{\text{true}})^\top\mathbf{o}\right|.$$

By the Cauchy-Schwarz inequality, $\left|(\mathbf{x})^T\mathbf{y}\right| \leq ||\mathbf{x}||_2\,||\mathbf{y}||_2$. Applying this:

$$|\mathbb{E}[Q|\mathbf{o}]_{\text{mis}} - \mathbb{E}[Q|\mathbf{o}]_{\text{true}}| \leq \frac{||\mathbf{k}_1||_2}{d_1}\left|\left|s \cdot \mathbf{v}_1 - \mathbf{u}_1^{\text{true}}\right|\right|_2||\mathbf{o}||_2$$

The term $||s \cdot \mathbf{v}_1 - \mathbf{u}_1^{\text{true}}||_2$ is equivalent to $||\mathbf{v}_1 - s \cdot \mathbf{u}_1^{\text{true}}||_2$ from the main theorem statement, where $s$ aligns $\mathbf{u}_1^{\text{true}}$ with $\mathbf{v}_1$. From the preceding Theorem, we have the bound (where $\delta = \frac{1}{d_1} ||\mathbf{k}_1||_2^2$):

$$||\mathbf{v}_1 - s \cdot \mathbf{u}_1^{\text{true}}||_2 \leq \frac{2 ||\mathbf{E}||_{\text{op}}}{\delta} = \frac{2 \left|\left|\sum_{\ell=2}^{h} \frac{1}{d_\ell} \mathbf{k}_\ell \mathbf{k}_\ell^T\right|\right|_{\text{op}}}{\frac{1}{d_1} ||\mathbf{k}_1||_2^2}$$

Substituting this bound into the inequality for the error in the conditional mean:

$$\begin{aligned}
|\mathbb{E}[Q|\mathbf{o}]_{\text{mis}} - \mathbb{E}[Q|\mathbf{o}]_{\text{true}}| &\leq \frac{||\mathbf{k}_1||_2}{d_1} \left( \frac{2 ||\mathbf{E}||_{\text{op}}}{\frac{1}{d_1} ||\mathbf{k}_1||_2^2} \right) ||\mathbf{o}||_2 \\
&= \frac{||\mathbf{k}_1||_2}{d_1} \cdot \frac{2 d_1 ||\mathbf{E}||_{\text{op}}}{||\mathbf{k}_1||_2^2} \cdot ||\mathbf{o}||_2 \\
&= \frac{2 ||\mathbf{E}||_{\text{op}}}{||\mathbf{k}_1||_2} ||\mathbf{o}||_2 \\
&= \frac{2 \left|\left|\sum_{\ell=2}^{h} \frac{1}{d_\ell} \mathbf{k}_\ell \mathbf{k}_\ell^T\right|\right|_{\text{op}}}{||\mathbf{k}_1||_2} ||\mathbf{o}||_2
\end{aligned}$$

$\square$

This corollary shows that the error in the estimated conditional mean of $Q$ (due to using the misspecified direction for $Q$'s influence) scales with:

- The magnitude of the observable vector $\mathbf{o}$ (specifically, $||\mathbf{o}||_2$).

- The collective strength of the confounding latent variables in the precision domain ($\left|\left|\sum_{\ell=2}^{h} \frac{1}{d_\ell} \mathbf{k}_\ell \mathbf{k}_\ell^T\right|\right|_{\text{op}}$).

- Inversely with the $\ell_2$-norm of the true connection weights of $Q$ ($||\mathbf{k}_1||_2$).

Especially, we see that strong confounders widen the gap bound, whereas heavier connection weights to the true score shrink it. Put differently, *misspecification hurts most when confounders are strong and the quality signal is weak.*

### D.5. Sample Complexity for CARE tensor algorithm

**Assumption D.8** (Model and identifiability). Let $J = (X_1^\top, X_2^\top, X_3^\top)^\top \in \mathbb{R}^p$ ($p = p_1 + p_2 + p_3$) be one observations i.i.d generated as

$$(Q, C) \sim \text{Multinomial}\big(\{\pi_{qc}\}_{q,c \in \{0,1\}}\big),$$
$$X_\ell \mid (Q = q, C = c) \sim \mathcal{N}\big(\mu_{qc}^{(\ell)}, \Sigma\big).$$

with $\ell \in \{1, 2, 3\}$. Write $r \in [4] \leftrightarrow (q, c) \in \{0, 1\}^2$ and define $w_r := \pi_{qc}$, $a_r := \mu_{qc}^{(1)} \in \mathbb{R}^{p_1}$, $b_r := \mu_{qc}^{(2)} \in \mathbb{R}^{p_2}$, $c_r := \mu_{qc}^{(3)} \in \mathbb{R}^{p_3}$.

(A1) **Block-conditional independence.** $X_1 \perp X_2 \perp X_3 \mid (Q, C)$.

(A2) **Full-rank moment tensor.** The population third-order moment $M := \mathbb{E}[X_1 \otimes X_2 \otimes X_3] = \sum_{r=1}^{4} w_r\, a_r \otimes b_r \otimes c_r$ has rank 4, with $\pi_{\min} := \min_r \pi_r > 0$ and $\lambda_{\min} := \min_r ||a_r||_2 ||b_r||_2 ||c_r||_2 > 0$.

(A3) **Non-degenerate covariance.** $\sigma_{\max}^2 := ||\Sigma||_{\text{op}} < \infty$.

(A4) **Spectral gap.** The CP factors are uniquely defined up to scaling/sign and satisfy the eigenvalue-gap condition of Theorem 5.1 in (Anandkumar et al., 2014). Denote that gap by $\delta > 0$.

(A5) **Correct graph partition.** There exist a graph partition such that judges between different groups are conditional independent. Step A of Algorithm 2 returns the true groups $\mathcal{G}_1, \mathcal{G}_2, \mathcal{G}_3$.

**Theorem D.9** (Sample complexity of CARE tensor step). *Fix $0 < \varepsilon < 1$ and let the assumptions above hold. Run Algorithm 2 (CARE) on $n$ i.i.d. samples to obtain $\{\hat{\mu}_{qc}, \hat{\pi}_{qc}\}_{q,c\in\{0,1\}}$. Under Assumption D.8, there exist universal constants $C_1, C_2 > 0$ such that if*

$$n \;\geq\; C_1 \,\frac{\sigma_{\max}^6}{\delta^2\,\pi_{\min}^2}\, p\, \log(p/\varepsilon),$$

*then with probability at least $1 - \varepsilon$*

$$\max_{q,c} \|\hat{\mu}_{qc} - \mu_{qc}\|_2 \leq C_1 \frac{\sigma_{\max}^3}{\delta} \sqrt{\frac{p\log(p/\varepsilon)}{n}},$$

$$\max_{q,c} |\hat{\pi}_{qc} - \pi_{qc}| \leq C_2 \sqrt{\frac{p\log(p/\varepsilon)}{n}}.$$

We defer the proof to D.6.

### D.6. Proofs

**Proof of Theorem D.3**

*Proof.* Let $K_{JH} = [\mathbf{k}_1, \ldots, \mathbf{k}_h]$ and $K_{HH}^{-1} = \mathrm{diag}(\lambda_1, \ldots, \lambda_h)$. Then

$$L = K_{JH} K_{HH}^{-1} K_{HJ} = \sum_{i=1}^h \lambda_i\, \mathbf{k}_i \mathbf{k}_i^\top.$$

Under Assumption D.2 strengthened to orthonormal columns, we have $\mathbf{k}_i^\top \mathbf{k}_j = \delta_{ij}$. Hence, for each $i$,

$$L\,\mathbf{k}_i = \sum_{j=1}^h \lambda_j\, \mathbf{k}_j (\mathbf{k}_j^\top \mathbf{k}_i) = \lambda_i\, \mathbf{k}_i,$$

so $(\lambda_i, \mathbf{k}_i)$ is an eigenpair of $L$. Because $\lambda_1 > \cdots > \lambda_h > 0$ are distinct, these eigenvectors are unique up to sign, and ordering is only up to permutation. Therefore the columns of $K_{JH}$ are identifiable from $L$ up to sign flips and permutations. $\square$

**Proof of Theorem D.4**

*Proof.* Write $\tilde{L} = L + \Delta$, where

$$\tilde{L} = (K_{JH} + E)\, K_{HH}^{-1}\, (K_{JH} + E)^\top$$
$$= K_{JH} K_{HH}^{-1} K_{HJ} + K_{JH} K_{HH}^{-1} E^\top + E K_{HH}^{-1} K_{HJ} + E K_{HH}^{-1} E^\top.$$

Thus

$$\Delta = K_{JH} K_{HH}^{-1} E^\top + E K_{HH}^{-1} K_{HJ} + E K_{HH}^{-1} E^\top.$$

By sub-multiplicativity of the operator norm,

$$\|\Delta\|_2 \leq 2\,\|K_{JH}\|_2\,\|K_{HH}^{-1}\|_2\,\|E\|_2 + \|K_{HH}^{-1}\|_2\,\|E\|_2^2.$$

Under the orthonormal-column assumption $K_{JH}^\top K_{JH} = I_h$, the singular values of $K_{JH}$ are all 1, hence $\|K_{JH}\|_2 = 1$. Therefore,

$$\|\Delta\|_2 \leq 2\,\|K_{HH}^{-1}\|_2\,\|E\|_2 + \|K_{HH}^{-1}\|_2\,\|E\|_2^2.$$

Applying the Davis–Kahan eigenvector perturbation theorem for symmetric matrices to the isolated eigenvalue $\lambda_i$ (gap $\delta_i$) yields that there exists $s_i \in \{\pm 1\}$ such that

$$\|\tilde{u}_i - s_i u_i\|_2 \leq \frac{2\|\Delta\|_2}{\delta_i}.$$

Substituting the bound on $\|\Delta\|_2$ gives

$$\|\tilde{u}_i - s_i u_i\|_2 \leq \frac{4\,\|K_{HH}^{-1}\|_2\,\|E\|_2}{\delta_i} + O(\|E\|_2^2),$$

which completes the proof. □

**Proof of Theorem D.5**

*Proof.* **Step 1 – Spectral error of $\hat{L}_n$.** Apply Chandrasekaran et al.'s Theorem 4.1 with the regularization parameters

$$\gamma_n = \frac{48\sqrt{2}\,D\psi(2-\nu)}{\xi(T)\nu}\sqrt{\frac{\epsilon}{n}},$$

$\sigma, \theta$ as in conditions (3)–(4).

Under the incoherence and curvature conditions of their Proposition 3.3, there exists a universal constant $C_1 > 0$ such that, with probability at least $1 - 2e^{-\epsilon}$,

$$\big\| \hat{L}_n - L^* \big\|_2 \; \leq \; C_1\,\frac{\sqrt{\epsilon/n}}{\xi(T)}. \tag{3}$$

**Step 2 – Eigenvector perturbation via Davis–Kahan.** Let $L^* = U\Lambda U^\top$ with $\Lambda = \mathrm{diag}(\lambda_1, \ldots, \lambda_h, 0, \ldots, 0)$ and collect the top–$h$ eigenvectors in $U_h = [u_1, \ldots, u_h]$. Write the spectral decomposition of the estimator as $\hat{L}_n = \hat{U}_h \hat{\Lambda} \hat{U}_h^\top + R$, where $R$ contains only the eigen-components of rank $> h$. Set the perturbation $E := \hat{L}_n - L^*$.

Applying Corollary 3 from (Yu et al., 2015) to the $i$-th eigenpair gives

$$\|u_i - \hat{u}_i\|_2 \; \leq \; \frac{2^{3/2}\|E\|_2}{\delta_i}. \tag{4}$$

**Step 3 – Combine the two bounds.** Insert equation 3 into equation 4:

$$\| \hat{u}_i - u_i \|_2 \; \leq \; \frac{2^{3/2}C_1}{\delta\,\xi(T)}\sqrt{\frac{\epsilon}{n}} \qquad \forall\, i \in [h],$$

and take the maximum over $i$. This proves the advertised high-probability bound

$$\max_{i \leq h} \| \hat{u}_i - u_i \|_2 \; = \; O\!\Big(\tfrac{\sqrt{\epsilon/n}}{\xi(T)\,\delta}\Big).$$

**Step 4 – Invert to a sample-size requirement.** Set the right-hand side to a target accuracy $\alpha \in (0,1)$, i.e.,

$$\frac{2^{3/2}C_1}{\xi(T)\,\delta}\sqrt{\frac{\eta}{n}} \leq \alpha.$$

Solving for $n$ yields

$$n \; \geq \; \frac{8C_1^2}{\xi(T)^2\,\delta^2} \cdot \frac{\eta}{\alpha^2}.$$

Equivalently, to ensure failure probability at most $\zeta \in (0,1)$, set $\eta = \log(2/\zeta)$. □

**Proof for Theorem D.9.**

*Proof.* Throughout the proof, we condition on that Step A of Algorithm 2 returns the true partition $(\mathcal{G}_1, \mathcal{G}_2, \mathcal{G}_3)$, which holds by Assumption D.8(A5). Hence each sample can be written as $J = (X_1^\top, X_2^\top, X_3^\top)^\top$ with $X_\ell \in \mathbb{R}^{p_\ell}$ and $p = p_1 + p_2 + p_3$.

We start by identifying population data structure. Recall the indexing $r \in [4] \leftrightarrow (q, c) \in \{0, 1\}^2$ and define $w_r = \pi_{qc}$, $a_r = \mu_{qc}^{(1)}$, $b_r = \mu_{qc}^{(2)}$, $c_r = \mu_{qc}^{(3)}$. By block-conditional independence (A1),

$$\mathbb{E}[X_1 \otimes X_2 \otimes X_3 \mid (Q, C) = r] = \mathbb{E}[X_1 \mid r] \otimes \mathbb{E}[X_2 \mid r] \otimes \mathbb{E}[X_3 \mid r] = a_r \otimes b_r \otimes c_r.$$

Taking expectation over $r$ yields

$$M := \mathbb{E}[X_1 \otimes X_2 \otimes X_3] = \sum_{r=1}^{4} w_r \, a_r \otimes b_r \otimes c_r. \tag{5}$$

Assumption D.8(A2) ensures this CP representation has rank 4 and each component is non-degenerate.

To bound the estimation error, we consider the empirical tensor:

$$\hat{M} := \frac{1}{n} \sum_{i=1}^{n} X_1^{(i)} \otimes X_2^{(i)} \otimes X_3^{(i)} \in \mathbb{R}^{p_1 \times p_2 \times p_3}.$$

Let the tensor operator norm be

$$\|T\|_{\text{op}} := \sup_{\|u\|_2 = \|v\|_2 = \|w\|_2 = 1} \langle T, \ u \otimes v \otimes w \rangle.$$

Fix unit vectors $(u, v, w)$ and set

$$Z_i(u, v, w) := \langle X_1^{(i)}, u \rangle \langle X_2^{(i)}, v \rangle \langle X_3^{(i)}, w \rangle.$$

Then

$$\langle \hat{M} - M, \ u \otimes v \otimes w \rangle = \frac{1}{n} \sum_{i=1}^{n} \Big( Z_i(u, v, w) - \mathbb{E} Z_i(u, v, w) \Big).$$

Under Assumption D.8(A3), each one-dimensional projection $\langle X_\ell^{(i)} - \mathbb{E}[X_\ell^{(i)} \mid (Q, C)], t \rangle$ is centered Gaussian with variance at most $\|\Sigma\|_{\text{op}} = \sigma_{\max}^2$, hence is sub-Gaussian with parameter $O(\sigma_{\max})$. Since (A1) gives conditional independence of $X_1, X_2, X_3$ given $(Q, C)$, a standard product-of-sub-Gaussians argument yields that $Z_i(u, v, w) - \mathbb{E} Z_i(u, v, w)$ is sub-exponential with parameter $O(\sigma_{\max}^3)$ uniformly over unit $(u, v, w)$. Therefore Bernstein's inequality implies that for universal constants $c, C > 0$,

$$\Pr\Big(\big|\langle \hat{M} - M, \ u \otimes v \otimes w \rangle\big| \geq t\Big) \leq 2 \exp\Big(-c \, n \min\{t^2/\sigma_{\max}^6, \ t/\sigma_{\max}^3\}\Big). \tag{6}$$

To upgrade from fixed $(u, v, w)$ to the supremum, take an $\eta$-net $\mathcal{N}_\ell$ of the unit sphere in $\mathbb{R}^{p_\ell}$ with $|\mathcal{N}_\ell| \leq (1 + 2/\eta)^{p_\ell}$. Choosing $\eta = 1/6$ gives $|\mathcal{N}_\ell| \leq 13^{p_\ell}$ and hence $|\mathcal{N}_1 \times \mathcal{N}_2 \times \mathcal{N}_3| \leq 13^p$. A standard net argument for multilinear forms yields

$$\|\hat{M} - M\|_{\text{op}} \leq 2 \max_{u \in \mathcal{N}_1, \ v \in \mathcal{N}_2, \ w \in \mathcal{N}_3} \big|\langle \hat{M} - M, \ u \otimes v \otimes w \rangle\big|. \tag{7}$$

Applying the union bound over the net in equation 7 and then equation 6 yields: for a sufficiently large universal constant $C > 0$, taking $t = C \sigma_{\max}^3 \sqrt{\frac{p \log(p/\varepsilon)}{n}}$ gives

$$\|\hat{M} - M\|_{\text{op}} \leq C \sigma_{\max}^3 \sqrt{\frac{p \log(p/\varepsilon)}{n}} \qquad \text{with probability at least } 1 - \varepsilon/2. \tag{8}$$

Given this concentration result, we treat $E := \hat{M} - M$ as the perturbation. By Assumption D.8(A4), the rank-4 CP decomposition equation 5 is identifiable (up to permutation/sign/scale) and moreover lies in the perturbation regime of the robust tensor power method analyzed by Anandkumar et al. (2014). Concretely, their perturbation theorem provides the following implication: there exist constants $c_0, C_{\text{dec}} > 0$ such that whenever

$$\|E\|_{\text{op}} \leq c_0 \, \delta, \tag{9}$$

the tensor power/deflation routine returns a set of 4 recovered components $\{(\hat{w}_r, \hat{a}_r, \hat{b}_r, \hat{c}_r)\}_{r=1}^4$ for which there exist a permutation $\tau$ of $[4]$ and signs $s_{r,\ell} \in \{\pm 1\}$ such that for all $r \in [4]$,

$$\|\hat{a}_{\tau(r)} - s_{r,1} a_r\|_2 \ \vee \ \|\hat{b}_{\tau(r)} - s_{r,2} b_r\|_2 \ \vee \ \|\hat{c}_{\tau(r)} - s_{r,3} c_r\|_2 \ \leq \ C_{\text{dec}} \frac{\|E\|_{\text{op}}}{\delta}, \tag{10}$$

and the recovered mixing weights satisfy a Lipschitz stability bound of the form

$$\max_{r \in [4]} |\hat{w}_{\tau(r)} - w_r| \ \leq \ C_\pi \|E\|_{\text{op}}, \tag{11}$$

for a constant $C_\pi > 0$ that depends only on the (fixed) rank 4 and the conditioning quantities implicit in Assumption D.8(A4).

We then aggregate these factors into the full mean vectors $\mu_r := \text{concat}(a_r, b_r, c_r) \in \mathbb{R}^p$ and $\hat{\mu}_{\tau(r)} := \text{concat}(\hat{a}_{\tau(r)}, \hat{b}_{\tau(r)}, \hat{c}_{\tau(r)})$. After aligning signs as in equation 10,

$$\|\hat{\mu}_{\tau(r)} - \mu_r\|_2^2 = \|\hat{a}_{\tau(r)} - a_r\|_2^2 + \|\hat{b}_{\tau(r)} - b_r\|_2^2 + \|\hat{c}_{\tau(r)} - c_r\|_2^2 \ \leq \ 3 C_{\text{dec}}^2 \frac{\|E\|_{\text{op}}^2}{\delta^2}.$$

Taking square roots and maximizing over $r$ gives

$$\max_{r \in [4]} \|\hat{\mu}_{\tau(r)} - \mu_r\|_2 \ \leq \ \sqrt{3} \, C_{\text{dec}} \frac{\|E\|_{\text{op}}}{\delta}. \tag{12}$$

Combining equation 12 with equation 8 yields

$$\max_{r \in [4]} \|\hat{\mu}_{\tau(r)} - \mu_r\|_2 \ \leq \ C_1 \frac{\sigma_{\max}^3}{\delta} \sqrt{\frac{p \log(p/\varepsilon)}{n}}$$

with probability at least $1 - \varepsilon/2$, after enlarging $C_1$ by an absolute factor. Translating $r \leftrightarrow (q,c)$ gives the first claimed bound.

Similarly, the recovered mixing weights satisfy the stability bound. By equation 11 and equation 8, with probability at least $1 - \varepsilon/2$,

$$\max_{r \in [4]} |\hat{w}_{\tau(r)} - w_r| \ \leq \ C_\pi \|E\|_{\text{op}} \ \leq \ C_\pi \, C \, \sigma_{\max}^3 \sqrt{\frac{p \log(p/\varepsilon)}{n}}.$$

Under the normalization implicit in Assumption D.8(A4) (which fixes the scaling ambiguity in CP factors), the above translates to the stated $O(\sqrt{p \log(p/\varepsilon)/n})$ rate for $\max_{q,c} |\hat{\pi}_{qc} - \pi_{qc}|$ (after absorbing model-independent fixed scalings into the constant $C_2$). Translating $r \leftrightarrow (q,c)$ gives the second bound.

Finally, it remains to ensure the decomposition operates in the perturbative regime equation 9. By equation 8, the sufficient condition $\|E\|_{\text{op}} \leq c_0 \delta$ holds whenever

$$C \, \sigma_{\max}^3 \sqrt{\frac{p \log(p/\varepsilon)}{n}} \ \leq \ c_0 \, \delta, \quad \text{i.e.,} \quad n \ \geq \ \frac{C^2}{c_0^2} \frac{\sigma_{\max}^6}{\delta^2} \, p \log(p/\varepsilon).$$

Incorporating the additional detectability requirement for the smallest component (Assumption D.8(A2) with $\pi_{\min} > 0$) yields the stated sufficient sample size condition $n \geq C_1 \sigma_{\max}^6 (\delta^2 \pi_{\min}^2)^{-1} p \log(p/\varepsilon)$. Finally, union-bounding the concentration event with the internal success probability of the randomized initialization in the tensor power/deflation routine yields the final success probability $1 - \varepsilon$. $\qquad\square$

# E. Experiment Details

In this section, we provide experimental details and additional experimental results. We describe datasets details, evaluation prompts we used to collect LLM judgments. In addition, we introduce the construction of programmatic judge and examples. Finally, we include additional synthetic experiments to show the robustness of our method in ideal settings.

## E.1. Datasets

We evaluate our methods on a diverse set of public benchmarks covering both scoring and pairwise preference tasks. In scoring datasets, LLM judges assign real-valued ratings whose range matches the human annotation scale of each dataset (e.g., 0–4, 1–5, 0–10, or 0–100). In preference datasets, LLM judges provide comparative ratings on a $-3, \ldots, 3$ scale, where the sign indicates which response is preferred and the magnitude reflects confidence in that preference.

**ASSET (Text Simplification) (Alva-Manchego et al., 2020).** A sentence simplification dataset where each original sentence is paired with multiple simplified versions, annotated by human raters across multiple aspects such as meaning preservation, fluency, and simplicity, each scored from 0 to 100. We use the `human_rating` field as the scalar target and evaluate aggregation quality against these human scores. The dataset is publicly available at `facebook/asset`.

**FeedbackQA (Li et al., 2022).** A question-answering dataset with human-provided scalar ratings of answer helpfulness, ranging from 1 to 5. We use the validation set in our experiments, treating the average of two human ratings as the ground truth. The dataset is publicly available at `McGill-NLP/feedbackQA`.

**Review-5K (Peer Review) (Weng et al., 2025).** The Review-5K dataset is a collection of peer reviews and associated metadata from the ICLR 2024 conference. Each review includes reviewer messages with guidelines and paper content, along with multiple aspect ratings such as clarity, originality, and soundness, all scored from 1 to 10 following ICLR's review format. We use the average of available ratings (`avg_rate`) as the ground-truth scalar. The dataset is publicly available at `WestlakeNLP/Review-5K`.

**Summarize-from-Feedback (Stiennon et al., 2020).** A large-scale summarization dataset introduced by OpenAI, containing human and model-written summaries with human feedback annotations. For scoring experiments, we use the `axis` split, which provides scalar ratings (0–7) assessing summary quality along multiple axes such as helpfulness, correctness, and coherence. For preference experiments, we use the `comparisons` split, which contains pairwise judgments indicating which of two summaries is preferred; these examples are taken from the `Summarize-from-Feedback` subset of the `allenai/preference-test-sets` (Lambert et al., 2024) collection. We randomly sample 5,000 examples for each setting. The original dataset is publicly available at `openai/summarize_from_feedback`.

**UltraFeedback (Cui et al., 2023).** A scalar feedback dataset where assistant responses are rated from 0 to 10 based on overall quality, using scores aggregated from GPT-4 and human raters. We randomly sample 5,000 examples for evaluation. The dataset is publicly available at `openbmb/UltraFeedback`.

**Yelp (Zhang et al., 2015)** A large-scale review dataset containing user-written reviews with 5-star ratings (1–5) as ground-truth labels, where higher scores indicate more positive sentiment. For fully-Gaussian evaluation, we randomly sample 5,000 examples from the test split with its numeric ratings as scalar targets. The dataset is publicly available at `Yelp/yelp_review_full`.

**Chatbot Arena Conversations (Zheng et al., 2023b)** A large-scale pairwise comparison dataset from LMSYS, where annotators select the better of two model responses to the same user prompt. We use the dataset's binary preference signal (`"A"` vs. `"B"`) as the ground truth. For evaluation, we randomly sample 5,000 examples. The dataset is publicly available at `lmsys/chatbot_arena_conversations`.

**CivilComments (Borkan et al., 2019)** We use a random subset of 5,000 examples from the CivilComments dataset (Borkan et al., 2019). Each comment is annotated with a continuous `toxicity` score between 0 and 1, representing the fraction of annotators who labeled it as toxic; for binary classification, we threshold this score at 0.5. In our setup, LLM judges are asked to assign real-valued toxicity ratings on a 0–9 scale to enable scalar aggregation analysis. The dataset is publicly available at `google/civil_comments`.

**PKU Preferences (Better/Safer, Pairwise) (Ji et al., 2024).** Two pairwise preference datasets reflecting judgments of which response is *overall better* or *safer*. We use the provided `"A"` vs. `"B"` preference labels as the binary ground truth for each split. These datasets are taken from the `PKU-BETTER` and `PKU-SAFER` subsets of the `allenai/preference-test-sets` (Lambert et al., 2024) collection.

**SHP (Stanford Human Preferences, Pairwise) (Ethayarajh et al., 2022).** A pairwise preference dataset containing prompts, two model responses, and a human-selected preferred response. We use the binary preference (`"A"` vs. `"B"`) as the ground-truth label for aggregation experiments. The dataset is taken from the `SHP` subset of the `allenai/preference-test-sets` (Lambert et al., 2024) collection.

**Humans–LLMs Judgment Bias Dataset (Section 5.4) (Chen et al., 2024).** A semi-synthetic benchmark introduced to analyze bias in LLM-as-a-judge evaluations. Each example includes an original model response and a stylistically perturbed variant crafted to induce a specific bias while preserving semantics. In our robustness experiments, we use only stylistic perturbations—*authority* and *beauty*—to test aggregation robustness against superficial confounders. The *authority bias* is simulated by adding fake citations to evoke a sense of credibility, e.g., "...(Weisstein, Eric W. 'Square Root.' *MathWorld*...)." The *beauty bias* is simulated by inserting emojis or decorative formatting, e.g., "$\boxed{6}$ multiplied by $\boxed{6}$ equals 36." The dataset is publicly available at `Humans_LLMs_Judgement_Bias`.

**Multi-subject RLVR (Section 5.5) (Zhao et al., 2025; Su et al., 2025).** A multi-domain QA corpus used in RL with verifiable rewards, adapted here to test judge robustness to "master-key" trigger attacks. Following the setup in Section 5.4, each item pairs the question and its reference answer with an *adversarially crafted, incorrect* response (e.g., minimal token triggers or reasoning openers). LLM judges are asked to assess correctness; because the adversarial response is wrong by construction, any "correct" judgment is a false positive. We report false-positive rates by attack type to evaluate CARE's effectiveness as a defense against trigger-based failures of LLM-as-a-judge.

## E.2. Computing Resources

We used a server equipped with an NVIDIA A100 (40GB). Generating LLM judge outputs took up to 3 hours per dataset.

## E.3. Hyperparameter Tuning

We reserve 15% of the data as a validation set. For CARE-SVD, we search over $\gamma_n \in \{0.1, 0.2, 0.25, 0.5, 0.75, 1, 2, 3, 5, 7, 10\}$ and select the value that yields the best performance on the validation set. For CARE-TENSOR, we perform a grid search over $\gamma_n \in \{10^{-3}, 5 \times 10^{-3}, 10^{-2}, 5 \times 10^{-2}, 10^{-1}\}$ and $\tau \in \{10^{-3}, 5 \times 10^{-3}, 10^{-2}, 5 \times 10^{-2}, 10^{-1}\}$, choosing the combination that achieves the best validation performance.

## E.4. LLM Judge Models

We use the following large language models (LLMs) as judges. For continuous scoring tasks, we employ `Llama-3.2-{1,3}B` (Meta, 2024), `Mistral-7B-Instruct-v0.3` (Mistral, 2023), `Qwen3-{0.6,1.7,4,8}B` (Alibaba, 2025), `Phi-4-mini-instruct` (Microsoft, 2025), and `Gemma-3-{1,4}B-IT` (DeepMind, 2025). For classification and preference tasks, we additionally include `Llama-3.2-{1,3}B-Instruct` (Meta, 2024), `Gemma-2-9B-IT` (DeepMind), `Qwen2.5-{0.6,1.7,4,8}B`, and `Yi-1.5-{6,9}B-Chat` (01.AI, 2024). In each dataset, we exclude any model whose outputs contain more than 10% missing values or out-of-range outputs.

## E.5. Prompt Templates

In this subsection we provide the prompts we used for collecting LLM judgements.

> **LLM Judge Scoring Template (ASSET – Sentence Simplification)**
>
> You are rating the quality of a simplified sentence for a specific aspect of text simplification. Examine both sentences carefully before assigning a single integer score in [0, 100]. Use the whole range; do not default to 50 or round to multiples of 5 unless warranted.

**Aspect to evaluate:** {aspect}
Interpret the aspect as follows (case-insensitive):

- **meaning / meaning preservation** — Judge how faithfully the simplification preserves the original meaning (facts, relations, intent). Ignore simplicity/fluency except where they alter meaning.
- **fluency / grammaticality** — Judge grammatical correctness and naturalness of the simplification. Ignore meaning differences and degree of simplification; do not reward added content.
- **simplicity / readability** — Judge how much easier to read the simplification is compared with the original (shorter sentences, simpler vocabulary/structure). Minor meaning loss may be acceptable, but removing essential information or making text ambiguous should not receive top scores.
- **other** — If the aspect is unrecognised, judge overall simplification quality, prioritising meaning preservation, then fluency, then simplicity.

**Anchor guidance (choose any integer within each band):**

- 0–20 Unusable for the aspect.
- 21–40 Poor for the aspect.
- 41–60 Mixed: noticeable problems but some positives.
- 61–80 Good: solid performance with minor issues.
- 81–95 Very good: strong performance with only tiny flaws.
- 96–100 Excellent for the aspect.

**General rules:**

- Focus strictly on the stated aspect when scoring.
- Do not reward hallucinations or added facts not in the original.
- Non-round integers are allowed (e.g., 73, 88) if they best reflect quality.

Respond strictly in this format:

```
Feedback:::
Total rating: X
```

Where `X` is an integer in [0, 100]. Output nothing else.
**Original sentence:** {original}
**Candidate simplification:** {simplification}
**Expected output:**

```
Feedback:::
Total rating:
```

---

### LLM Judge Scoring Template (FeedbackQA, Ultrafeedback)

You will be given a user_question and system_answer couple.

Your task is to provide a 'total rating' scoring how well the system_answer answers the user concerns expressed in the user_question. Give your answer as a float on a scale of {min_rating} to {max_rating}, where {min_rating} means that the system_answer is not helpful at all, and {max_rating} means that the answer completely and helpfully addresses the question.

Provide your feedback as follows:

Feedback:::
Total rating: (your rating, as a float between {min_rating} and {max_rating})

Now here are the question and answer.

Question: {question}
Answer: {answer}

Feedback:::
Total rating:

---

## LLM Judge Scoring Template (Review-5K – Research Paper Review Quality)

You are scoring the **overall quality of a research paper submission**. Use the reviewing guidelines below to assess methodological soundness, clarity, contribution, and overall merit. Assign one of the standard conference rating levels (an integer in [1, 10]) according to this rubric:

- 1 – **Strong Reject**: fundamental flaws; clearly unsuitable for publication.
- 3 – **Reject, not good enough**: major weaknesses outweigh contributions.
- 5 – **Marginally below acceptance**: borderline but leaning negative.
- 6 – **Marginally above acceptance**: borderline but leaning positive; improvements still desirable.
- 8 – **Accept, good paper**: solid contribution with meaningful advances and only minor issues.
- 10 – **Strong Accept**: outstanding paper with exceptional clarity and impact.

If you believe an intermediate score (2, 4, 7, 9) better reflects the work, you may use it, positioning it between the adjacent rubric descriptions above.
**Reviewing guidelines:** guidelines
Respond strictly in this format:

```
Feedback:::
Total rating: X
```

Where `X` is an integer between 1 and 10 (inclusive). Output nothing else.
**Input:**

```
Paper content:
{paper}

Feedback:::
Total rating:
```

---

## LLM Judge Scoring Template (Summarize-from-Feedback – Overall Quality)

You are judging the overall **quality and faithfulness** of a candidate summary generated from a longer text. Your task is to evaluate how well the summary captures the main ideas, factual details, and intent of the original passage, balancing **accuracy, coverage, and clarity**.
**Scoring criteria:**

- Prioritise **faithfulness**: summaries must not introduce false or unrelated content.
- Assess **coverage**: the main points, outcomes, and reasoning from the original text should be reflected.
- Evaluate **clarity and conciseness**: good summaries are readable and direct, without unnecessary details.
- Do not reward stylistic flourish or length; correctness and substance matter most.

Assign a single integer score from 1 to 7 based on the following rubric:

- 1 – Very poor: inaccurate or largely off-topic; major facts are wrong or missing.
- 2 – Poor: significant factual or coverage errors; misleading or incomplete.
- 3 – Fair: captures some ideas but omits key information or introduces confusion.

- 4 – Adequate: mostly correct but has gaps, minor hallucinations, or extraneous details.
- 5 – Good: covers key ideas faithfully with small omissions or slight verbosity.
- 6 – Very good: accurate, coherent, and clear with only minor imperfections.
- 7 – Excellent: fully faithful, comprehensive, and well-written.

**Guidance:**

- Focus on factual alignment and coverage, not stylistic preferences.
- Small rephrasings or wording changes should not affect the rating.
- Reserve 7 for summaries that are both faithful and complete.

Respond exactly in this format:

```
Feedback:::
Total rating: X
```

Where X is an integer from 1 to 7 (inclusive). Output nothing else.
**Inputs:**

```
Original text:
{original}

Summary:
{summary}

Feedback:::
Total rating:
```

## LLM Judge Scoring Template (Yelp – Restaurant Review Sentiment)

You are evaluating the **overall sentiment** of a Yelp restaurant review. Read the entire review, consider the reviewer's tone, described experiences, and final impression, then assign a single star rating from 1 to 5 according to this rubric:

- 1 – Strongly negative; serious problems or clear dissatisfaction.
- 2 – Mostly negative; several complaints outweigh positives.
- 3 – Mixed or neutral; balanced feedback with no strong lean.
- 4 – Mostly positive; satisfied with only minor issues.
- 5 – Highly positive; enthusiastic praise and strong recommendation.

Base the rating on the **reviewer's intent**, not your personal opinion, and ignore formatting quirks.
Respond exactly in this format:

```
Feedback:::
Total rating: X
```

Where X is an integer from 1 to 5 and nothing else.
**Input:**

```
Review:
{text}

Feedback:::
Total rating:
```

## LLM Judge Preference Template (No Tie)

You will compare two assistant responses (A and B) for the same user question. Judge which response better helps the user overall, considering **factual accuracy, completeness, clarity, safety, and tone**.

Return a single integer chosen from 0, 1, 2, 4, 5, 6 using this rubric:

- 0 : Response A is **far superior** to Response B
- 1 : Response A is **clearly better** than Response B
- 2 : Response A is **slightly better** than Response B
- 4 : Response B is **slightly better** than Response A
- 5 : Response B is **clearly better** than Response A
- 6 : Response B is **far superior** to Response A

**Important (must follow all points below or the answer is invalid):**

- Ties are **NOT allowed**. Never output 3 or any number outside 0, 1, 2, 4, 5, 6.
- If Response A is better overall, output only one of 0, 1, 2.
- If Response B is better overall, output only one of 4, 5, 6.
- If the responses are very close, choose 2 (if A is slightly better) or 4 (if B is slightly better).
- Output only the integer — no explanations, words, or additional text.

**Reminder:** Response A better $\Rightarrow$ 0/1/2 Response B better $\Rightarrow$ 4/5/6 Outputs outside these sets will be rejected. Respond strictly in this format:

```
Feedback:::
Total rating: X
```

Where X is one of 0, 1, 2, 4, 5, 6.
**Inputs:**

```
Question: {question}
Response A: {answer_a}
Response B: {answer_b}
Feedback:::
Total rating:
```

**Note.** For postprocessing and aggregation, this 6-point asymmetric scale is shifted to the range [-3, 3]. We adopted this transformation because LLM judges tend to be systematically biased toward positive values.

---

## LLM Judge Scoring Template (CivilComments – Toxicity Rating)

You are a **toxicity rater**. Given a single user comment, assign a discrete toxicity score on a 0–9 scale. Higher numbers indicate more toxic language.
**Return exactly one integer using this rubric:**

- 0 : Clearly non-toxic or benign language.
- 1 : Mostly neutral with only faint negativity.
- 2 : Slightly negative or dismissive language.
- 3 : Mild insults or disrespectful tone.
- 4 : Noticeable harassment, repeated negativity, or coarse language.
- 5 : Borderline toxic; sustained insults or hostility.
- 6 : Clearly toxic with direct insults or slurs.
- 7 : Highly toxic; explicit harassment or hateful language.
- 8 : Severe toxicity including threats or strong hate speech.
- 9 : Extremely toxic content with explicit threats or violent hate speech.

**Guidelines:**

- Focus on the **literal wording** of the comment.
- Penalize **hate, harassment, threats**, or extreme vulgarity.
- If the comment mixes toxic and mitigating language, choose the score that reflects the overall impact.

Respond strictly in this format:

```
Feedback:::
Total rating: X
```

Where `X` is an integer between 0 and 9 (inclusive). Output nothing else.
**Input:**

```
Comment: {text}

Feedback:::
Total rating:
```

### E.6. Programmatic Judges

Programmatic judges, synthesized by LLMs, distill the evaluation logic of LLMs into interpretable and cost-effective program code (Huang et al., 2025; Huang et al.). These programmatic judges provide specialized and independent assessments, offering an alternative to using LLMs directly as evaluators. Although cheap to obtain, they may inherently encode the biases or noise present in the originating LLMs. We incorporate these judges to evaluate CARE's aggregation effectiveness under conditions with increased signal noise.

We describe the creation of programmatic judges and the criteria we consider. We use OpenAI's GPT-4o (Hurst et al., 2024) to generate programmatic judges with the following prompt:

---

**Programmatic Judge Template**

**Task:** Create a Python function that serves as a judge to evaluate LLM-generated responses.
**Requirements:**

1. Function name: `program_judge`

2. Function signature: `program_judge(query: str, response: str) -> dict`

3. Return a dictionary with the following keys:

    - `'score'` (float, 0–10)
    - `'reasoning'` (string)
    - `'criteria'` (string)

4. The score ranges from 0 (lowest quality) to 10 (highest quality)

5. Higher scores indicate higher-quality responses

6. Implement a unique judging strategy

7. Include comprehensive error handling

8. Add a concise docstring describing the judging logic

**Instruction:** The judge may rely on third-party specialized models or Python libraries. Do not use Google search or web crawling. Generate a complete, unique judge function that assigns higher scores to higher-quality responses.

---

We follow the suggested criteria from Huang et al. (2025), which are studied that can be translated into executable code. Specifically, we synthesize 30 programmatic judges and progressively integrate them into our 10 LLM-based evaluators. The considered evaluation criteria include: (i) factual accuracy, (ii) logical coherence, (iii) clarity and conciseness, (iv) completeness or coverage of the answer, (v) relevance to the query (semantic similarity), (vi) language quality and readability, (vii) bias detection, (viii) safety and toxicity, and (ix) response verbosity or redundancy. For example:

- **Coherence**: The judge evaluates logical coherence by checking whether entities mentioned in the query also appear in the response. Greater overlap indicates stronger consistency (see Program 2).

- **Relevance**: A judge uses TF-IDF to convert questions and responses into word vectors, computing cosine similarity to

measure semantic alignment (see Program 3).

- **Readability**: A judge leverages a third-party API to evaluate language complexity, using metrics like the Flesch–Kincaid grade level (see Program 1).

All the used judging logic, conditions, and pre-defined keyword lists are generated by the LLM.

Here we provide several examples in our selected programs in Figure 4 to illustrate this approach.

*Listing 1.* Language Quality/Readability.

```python
import textstat

def program_judge(query: str, response: str) -> dict:

    # Use Flesch Reading Ease score to evaluate readability. Higher scores
        indicate easier readability.

    score = textstat.flesch_reading_ease(response)

    # Normalize the score to a 0-10 scale (original scale is 0-100)
    normalized_score = score / 10

    # Reasoning includes the original FRE score for transparency
    reasoning = f"The response received a Flesch Reading Ease score of {score},
        indicating its readability. Higher scores mean the text is easier to read
        ."

    return {'score': normalized_score, 'reasoning': reasoning, 'criteria': "
        Language Quality/Readability"}
```

*Listing 2.* Logical Coherence.

```python
import spacy
nlp = spacy.load("en_core_web_sm")

def program_judge(query: str, response: str) -> dict:
    """
    Judges responses based on Logical Coherence.
    Checks if entities in the query also appear in the response.
    """
    q_ents = [ent.text for ent in nlp(query).ents]
    r_ents = [ent.text for ent in nlp(response).ents]
    matched = [e for e in q_ents if e in r_ents]

    score = (len(matched) / len(q_ents) * 10) if q_ents else 10.0
    reasoning = f"The response contained {len(matched)} out of {len(q_ents)}
        important entities from the query."

    return {'score': score, 'reasoning': reasoning, 'criteria': "Logical
        Coherence"}
```

*Listing 3.* Completeness/Coverage of Answer.

```python
from sklearn.feature_extraction.text import TfidfVectorizer
from sklearn.metrics.pairwise import cosine_similarity
```

*Table 6.* Judge loadings on latent factors in CARE-SVD. Factor 1 corresponds to true quality $Q$; Factor 2 reflects a length confounder.

| Judge | $Q$ (true quality) | Length confounder |
|---|---|---|
| Qwen3-8B | 0.396 | -0.240 |
| Llama-3.1-8B-Instruct | 0.664 | -0.076 |
| gemma-3-4b-it | 0.706 | -0.152 |
| Llama-3.2-1B | -0.009 | -0.140 |
| Qwen3-4B | 0.180 | 0.008 |
| gemma-3-1b-it | 0.243 | 0.595 |
| Llama-3.2-3B | 0.033 | 0.057 |
| Phi-4-mini-instruct | 0.715 | -0.051 |
| Qwen3-1.7B | 0.199 | -0.012 |
| Mistral-7B-Instruct-v0.3 | 0.804 | 0.016 |
| dummy_eval_1 | 0.098 | 0.742 |
| dummy_eval_2 | 0.035 | 0.290 |
| human_eval_1 | 0.337 | 0.078 |
| human_eval_2 | 0.338 | 0.059 |

```
def program_judge(query: str, response: str) -> dict:
    """
    Judges responses based on completeness and coverage of the query using TF-IDF
        and cosine similarity.
    """
    tfidf = TfidfVectorizer()
    tfidf_matrix = tfidf.fit_transform([query, response])
    similarity = cosine_similarity(tfidf_matrix[0:1], tfidf_matrix[1:2])[0][0]

    score = similarity * 10
    reasoning = f"The response covers {similarity*100:.2f}% of the query."

    return {'score': score, 'reasoning': reasoning, 'criteria': "Completeness/
        Coverage of Answer"}
```

### E.7. Additional Controlled Experiment on Confounding Factors

Unlike the semi-synthetic perturbations in Section 5.4, here we investigate whether CARE can separate the true quality latent factor from naturally arising confounders in a more controlled setting. Specifically, we introduce two dummy judges whose scores are directly correlated with response length or the presence of specific words. If CARE functions as intended, CARE should recover a factor structure in which high-quality judges align with the true quality factor $Q$, while the dummy judges align with a distinct confounder.

**Setup.** We ran CARE-SVD on FEEDBACKQA with 14 judges: 10 LLM judges, 2 programmatic "dummy" judges (length/keyword-sensitive), and 2 human annotators. Factor loadings are reported in Table 6.

**Results.** The observed loadings align with our hypothesis:

- **Factor 1 (true quality $Q$).** This factor exhibits *broad, balanced loadings* across competent LLM judges and the two human judges, with much weaker loadings for the programmatic dummy judges. Within model families, larger models have higher loadings (e.g., Llama-3.1-8B > Llama-3.2-3B $\approx$ Llama-3.2-1B), suggesting that $Q$ reflects underlying capability. Instruction-tuned models (Mistral-7B-Instruct, Phi-4-mini-instruct, Llama-3.1-8B-Instruct, Gemma-3-4B-it) also show above-median loadings, consistent with their alignment to human rubrics.
- **Factor 2 (length confounder).** This factor is dominated by a *high, concentrated loading* on the length-sensitive dummy_eval_1, with a secondary loading on gemma-3-1b-it (0.59). In contrast, nearly all other judges—including both humans and stronger instruction-tuned models—have near-zero loadings. Such a one-sided, few-judge pattern is characteristic of a confounder rather than true quality.

*Table 7.* Aggregation performance across different scoring datasets, measured by MAE ($\downarrow$; mean $\pm$ std).

|  | ASSET | FeedbackQA | Review-5K | Summarize | UltraFeedback | Yelp |
|---|---|---|---|---|---|---|
| **1st Factor** | **$27.148 \pm 0.133$** | **$0.753 \pm 0.003$** | **$1.950 \pm 0.006$** | **$1.325 \pm 0.003$** | **$0.622 \pm 0.006$** | **$0.694 \pm 0.005$** |
| 2nd Factor | $31.757 \pm 0.100$ | – | $2.222 \pm 0.007$ | – | $0.781 \pm 0.045$ | $1.067 \pm 0.062$ |
| 3rd Factor | $32.399 \pm 0.159$ | – | $3.115 \pm 0.006$ | – | $0.939 \pm 0.010$ | $0.955 \pm 0.023$ |
| 4th Factor | $30.529 \pm 0.144$ | – | $2.667 \pm 0.004$ | – | $0.996 \pm 0.005$ | $1.125 \pm 0.012$ |
| 5th Factor | $31.807 \pm 0.170$ | – | $3.369 \pm 0.004$ | – | $1.040 \pm 0.062$ | $1.099 \pm 0.003$ |

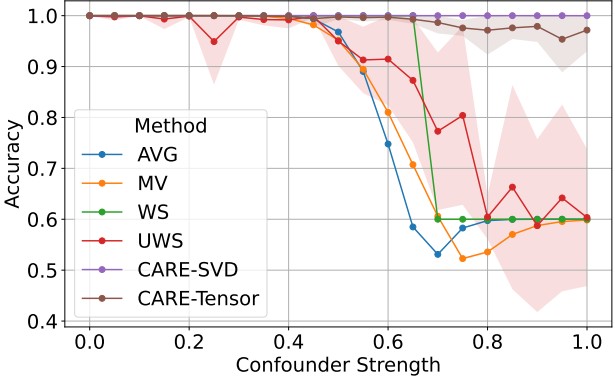

*(a)* **Second-order-sufficient.** CARE-SVD and CARE-Tensor remain robust as confounding increases, while baselines collapse.

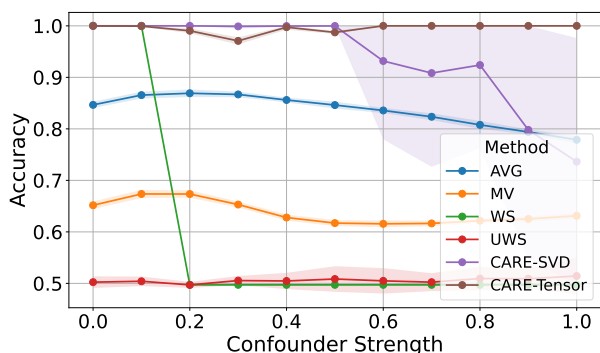

*(b)* **Second-order-insufficient.** Baselines and CARE-SVD degrade toward chance, while CARE-Tensor remains robust.

*Figure 6.* **Two synthetic regimes under latent confounding.** (a) In a second-order-sufficient regime, both CARE-SVD and CARE-Tensor remain accurate across the confounder-strength sweep, whereas standard baselines (AVG, MV, WS, UWS) deteriorate sharply at high confounding. (b) In a second-order-insufficient regime where $C \perp Q$ and second-order structure is confounder-dominated, baselines and CARE-SVD approach chance as confounding increases, while CARE-Tensor remains accurate by exploiting identifiable three-view third-order cross-moments.

### E.8. Ablation on Quality-Factor Selection in CARE-SVD

We validate the effectiveness of the heuristic that chooses the leading eigenvector as the quality factor in CARE-SVD.

**Setup.** We use the same scoring-task setup as in Table 1. Beyond the default heuristic (choosing the first factor), we also evaluate all recovered latent factors with non-negligible eigenvalues (eigenvalue $> 10^{-8}$). For each dataset, we treat each factor in turn as the quality factor, construct the corresponding aggregation rule, and report its test-set MAE. This allows us to quantify how close the default choice is to the best-performing factor in hindsight.[1]

**Results.** Table 7 shows that the leading factor consistently yields the best MAE among all recovered factors, supporting our default choice of the top eigenvector as the quality direction in CARE-SVD. We also observe that on datasets where higher-order factors are effectively negligible (e.g., FeedbackQA and Summarize), alternative factors are not meaningfully recoverable, and the benefit of CARE-SVD is correspondingly limited.

### E.9. Synthetic Experiments with Latent Confounding Factor

To evaluate the robustness of CARE under latent confounding, we report two synthetic regimes with increasing confounder strength. The first is a *second-order-sufficient* regime, where the dominant shared second-order structure remains aligned with the true-quality factor $Q$, so both CARE-SVD and CARE-Tensor remain robust as confounding increases. The second is a *second-order-insufficient* regime, where pairwise structure is dominated by a confounder $C$ (with $C \perp Q$), causing spectral recovery to degrade toward chance, while multi-view third-order moments remain identifiable. Across both regimes, CARE degrades substantially more gracefully than standard aggregation baselines as confounding strengthens.

---

[1]The "best factor" uses the test set for selection and is therefore optimistic; the goal is diagnostic rather than a deployable selection rule.

**Second-order-sufficient regime.**

**Setup.** We synthesize a binary true-quality label ($Q$) and a binary confounder ($C$), evaluated by five simulated judges under a Gaussian mixture model. Each view has a $4 \times 5$ conditional-mean table over the four $(C, Q)$ states. One view is primarily $Q$-driven, while the other two encode strong but oppositely signed confounder effects. We vary confounder strength $g \in [0, 1]$ via linear interpolation:

$$\tilde{\mu}_{(1,q)}(g) := \mu_{(0,q)} + g\big(\mu_{(1,q)} - \mu_{(0,q)}\big), \qquad q \in \{0, 1\},$$

where $g = 0$ removes the confounder and $g = 1$ restores its full effect. We generate 50,000 i.i.d. samples by sampling $(C, Q)$ with probabilities $(0.2, 0.3, 0.3, 0.2)$, drawing each view from $\mathcal{N}(\mu_{(C,Q)}, 0.01I)$, and concatenating the three views. Across 10 random seeds, we sweep $g \in \{0.0, 0.05, \ldots, 1.0\}$ and evaluate aggregation methods for recovering $Q$.

**Results.** Figure 6a shows that CARE-SVD and CARE-Tensor maintain high accuracy across the entire sweep, whereas baselines (AVG, MV, WS, UWS) degrade markedly as confounding strengthens.

**Second-order-insufficient regime.**

**Setup.** We sample $(C, Q)$ uniformly with $C \perp Q$ (each state has probability 0.25). Each sample has three views, each with $d = 12$ continuous judge features (total dimension 36), drawn from

$$X^{(v)} \mid (C, Q) \sim \mathcal{N}\big(\mu^{(v)}_{(C,Q)}, 0.01I\big), \qquad v \in \{1, 2, 3\}.$$

Within each view, the first $q_{\text{only}} = 3$ features are $Q$-only, the next $c_{\text{only}} = 8$ are $C$-only, and the remaining feature depends on both. We sweep a confounder strength $c \in [0, 1]$ by scaling only the confounder-dependent mean shifts:

$$\tilde{\mu}^{(v)}_{(1,q)}(c) := \mu^{(v)}_{(0,q)} + c\big(\mu^{(v)}_{(1,q)} - \mu^{(v)}_{(0,q)}\big), \qquad q \in \{0, 1\}.$$

Because most features are $C$-only, increasing $c$ makes the dominant second-order structure align with $C$, so CARE-SVD's leading low-rank direction becomes uninformative for predicting $Q$. In contrast, CARE-Tensor remains identifiable from three-view third-order cross-moments; we map recovered components to $Q$ using a small validation split (and use oracle tri-view grouping plus cross-view alignment to isolate the tensor step). For each seed, we sample 3,000 examples and evaluate methods over $c \in \{0.0, 0.1, \ldots, 1.0\}$ We report mean $\pm$ standard deviation over 25 seeds.

**Results.** Figure 6b shows that as confounding strengthens, baselines and CARE-SVD degrade toward chance accuracy, while CARE-Tensor remains robust across the sweep.

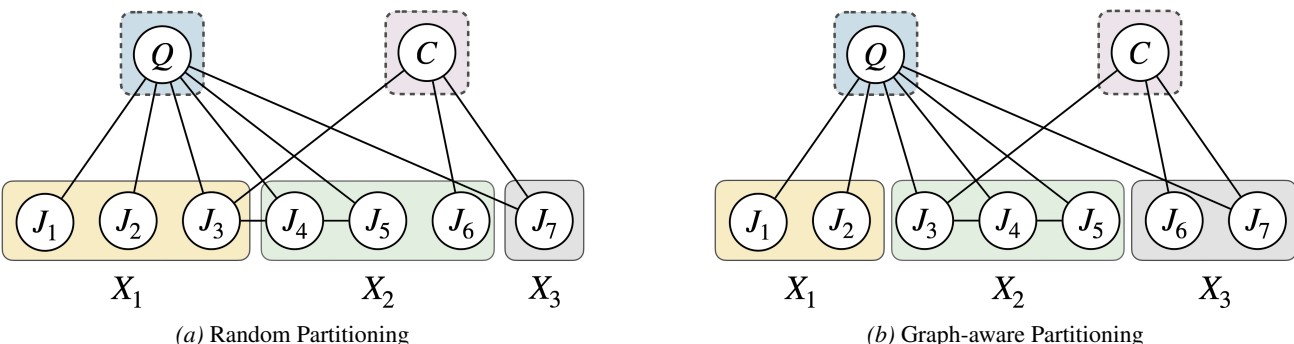

*(a)* Random Partitioning          *(b)* Graph-aware Partitioning

*Figure 7.* Random Partitioning vs. Graph Aware Partitioning. A random partitioning (a) leaves cross-view edges that violate the independence assumptions of tensor methods, whereas the graph-aware partitioning (b) considers cross-view edges and restores the required separation.

### E.10. Synthetic Experiment on Graph-Aware Tensor Decomposition

Tensor decomposition relies on a multi-view assumption: the judge groups (views) should be (approximately) conditionally independent given the latent variables. When judges exhibit conditional dependencies, a naive (random) split of judges into views can create many cross-view edges, violating this assumption and degrading tensor recovery. A key component of CARE-Tensor is to form views *using* the learned dependency structure, so as to minimize cross-view interactions. In this experiment, we evaluate whether this graph-aware view formation improves estimation accuracy.

*Figure 8.* $\ell_2$ reconstruction error (mean $\pm$ SD) for random vs. graph-aware grouping.

**Setup.** We simulated $n = 10{,}000$ items scored by $p = 12$ judges and partitioned the judges into three views of four judges each. To induce conditional dependencies, we planted within-view edges of strength $0.3$ at $40\%$ density (and no edges across the true views). We compared two view-formation strategies over ten random seeds (Figure 7):

- **Random partitioning:** assign judges to views uniformly at random, ignoring the dependency structure.

- **Graph-aware partitioning:** assign judges to views to minimize cross-block edges in the empirical precision matrix, aligning views with the learned conditional dependency structure.

Performance is measured by the $\ell_2$ error in recovering latent component means, i.e., $||\mu_{qc} - \hat{\mu}_{qc}||_2$.

**Results.** As shown in Figure 8, graph-aware partitioning reduces reconstruction error by more than an order of magnitude compared to random partitioning. This shows that **respecting judge dependencies during view formation is critical for accurate tensor recovery**, validating graph-aware view formation as a key component of CARE-Tensor.

