# OpenReview forum: "CARE: Confounder-Aware Aggregation for Reliable LLM Evaluation"
_ICML.cc/2026/Conference — ICML 2026 regular_

### Official Review · Reviewer_RWpU · 2026-03-04

**Soundness:** 3
**Presentation:** 3
**Significance:** 3
**Originality:** 3
**Overall Recommendation:** 4
**Confidence:** 4

**Summary:**

This paper addresses the problem of aggregating scores from multiple LLM judges that share systematic biases such as verbosity preference and positional bias. The authors propose CARE, a framework that decomposes the precision matrix of judge scores into sparse and low-rank components to separate direct judge dependencies from shared latent factors. Two estimators are provided: CARE-SVD for continuous scores under a Gaussian assumption, and CARE-Tensor for binary/mixture settings using CP tensor decomposition. Theoretical results cover identifiability, finite-sample recovery, stability, and misspecification bias. Experiments on 12 benchmarks show consistent improvements over standard aggregation methods, with up to 26.8% error reduction.

**Compliance With Llm Reviewing Policy:**

Affirmed.

**Key Questions For Authors:**

1. How often does the leading-eigenvector heuristic correctly identify the quality factor across the 12 benchmarks, and can this be validated without ground-truth labels?

2. Can you report the actual three-way judge partitions discovered for CARE-Tensor and whether they are stable across random seeds?

3. Can you empirically measure the misspecification bias on real benchmarks by comparing rank-1 and full CARE models?

4. How sensitive is CARE-SVD to the choice of k, and is there a principled way to select it from data?

5. Why is CARE-SVD absent from the adversarial defense experiment, and does it also provide robustness?

**Limitations:**

The paper acknowledges computational overhead and decomposition quality, but several important limitations are under-discussed: the leading-eigenvector heuristic could silently fail if a confounder dominates shared variation; the three-group requirement for CARE-Tensor and its practical fragility; the lack of guidance on selecting k; and the absence of statistical significance testing for the main improvement claims.

**Strengths And Weaknesses:**

**Strengths:**

**S1: The problem formulation is well-motivated and the modeling choice is principled.**
The sparse-plus-low-rank decomposition of the precision matrix is clean and well-justified, naturally connecting to latent-variable MRFs. The paper clearly articulates why existing methods that assume independence or model only annotator-specific noise are limited when judges share confounders.

**S2: The theoretical analysis is substantial and carefully structured.**
The identifiability analysis, finite-sample bounds, and misspecification analysis are nontrivial contributions for this application area. The misspecification result directly justifies the paper's premise by showing bias scales with confounder strength relative to the quality signal. The stability result under approximate orthogonality is also practically useful.

**S3: The breadth of experimental evaluation is a clear strength.**
Twelve benchmarks spanning continuous scoring, binary classification, and summarization quality demonstrate breadth. The consistent improvement pattern, with CARE-SVD winning on continuous tasks and CARE-Tensor on discrete/preference tasks, is convincing. The programmatic judge and adversarial defense experiments add practical value.

**S4: The confounder interpretability analysis is a nice touch.**
Showing that learned confounders correlate with human-interpretable features like verbosity and complexity provides evidence that the model captures semantically meaningful structure rather than noise.

**Weaknesses:**

**W1: The symmetry-breaking heuristics for identifying the quality factor are a significant weak point that undermines the theoretical guarantees.**
The theory proves identifiability up to sign and permutation, but deciding which direction is quality versus confounder relies on heuristics like "take the leading eigenvector." If a confounder like verbosity bias induces more cross-judge correlation than quality, this heuristic fails. Fallback heuristics exist in the appendix, but no systematic analysis shows how often the default heuristic fails or how sensitive results are to this choice.

**W2: The orthogonality assumption between quality and confounder loadings is strong and likely violated in practice.**
Exact recovery requires orthonormal loading columns, but if verbosity bias and quality both affect all judges, substantial overlap is inevitable. The approximate orthogonality heuristic helps in the non-orthogonal case but hurts in the orthogonal case, meaning practitioners need structural knowledge the method claims to avoid. The stability bound degrades as 1/delta_i, so near-collinear directions could yield poor recovery.

**W3: CARE-Tensor requires exactly three conditionally independent judge groups, which is a restrictive structural requirement.**
If the judge pool lacks a clean three-way partition, the method cannot be applied. With only around 10 to 19 judges in practice, finding three genuinely independent groups from an estimated sparse graph seems fragile. No sensitivity analysis on the tolerance parameter epsilon or discussion of how often this condition is met across benchmarks is provided.

**W4: The comparison to weak supervision baselines may be somewhat unfair given the different modeling assumptions.**
WS and UWS are general-purpose methods not designed for CARE's specific confounder structure. More natural competitors like PoLL, GED, and JudgeBlender are discussed in the related work or appendix but absent from the experimental comparison.

**W5: The sample complexity requirements for CARE-Tensor may be prohibitive for realistic evaluation settings.**
The theoretical bound could demand thousands of items, yet many LLM evaluation benchmarks have only a few hundred. The paper does not discuss whether the theoretical threshold is met in practice or show how performance degrades with smaller sample sizes.

**W6: The presentation of CARE-Tensor is compressed to the point of being hard to follow in the main text.**
Roughly half a page covers CARE-Tensor while the full algorithm is deferred to page 16 of the paper (in the appendix). For two co-equal contributions, this asymmetric treatment makes evaluating the tensor path difficult without extensive appendix-reading.

---

> ### Author Rebuttal · Authors · 2026-03-31
>
> We thank the reviewer for highlighting the problem formulation, the theoretical analysis, the breadth of the empirical evaluation, and the confounder-interpretability results.
>
> **On Q1 / W1: quality-factor identification and the leading-eigenvector heuristic.**
> We ran an additional CARE-SVD diagnostic: for each scoring dataset and seed, we scored each positive-eigenvalue factor in isolation by MAE against human scores and report the mean rank of the default leading factor (lower is better):
>
> | Scoring dataset | Mean rank of leading factor by MAE |
> |---|---:|
> | ASSET | 1.0 |
> | FeedbackQA | 1.0 |
> | Review-5K | 3.7 |
> | Summarize-from-Feedback | 1.0 |
> | UltraFeedback | 1.0 |
> | Yelp | 1.0 |
>
> Thus, on five of six scoring benchmarks, the leading factor is essentially rank-1 on average. Review-5K is the main exception, but alternative factor choices remain in a similar MAE range (best-anchor mean 1.906 vs. Table 1 CARE-SVD 1.957), suggesting entangled quality/confounding directions rather than estimator failure. Exact validation without labels is impossible, so we use leading-eigenvector selection as a heuristic supplemented by anchor and loading-balance diagnostics.
>
> **On Q2 / W3: CARE-Tensor group discovery and partition stability.**
> We also ran a 5-seed partition-stability analysis for CARE-Tensor:
>
> | Dataset | Stability across 5 seeds |
> |---|---|
> | PKU-BETTER | same partition in 5/5 |
> | PKU-SAFER | same partition in 5/5 |
> | CivilComments | dominant partition in 4/5 |
> | SHP | dominant partition in 4/5 |
> | Summarize | dominant partition in 4/5 |
> | Chatbot-Arena | 3 distinct partitions across 5 seeds |
>
> These results suggest that the learned grouping is often stable rather than arbitrary.
>
> **On W2: non-orthogonality.**
> **The orthogonality assumption is an identifiability condition**, not a claim that real judge panels are exactly orthogonal. Section 3 analyzes stability under approximate orthogonality, and Appendix D.4 gives an explicit misspecification bound showing how error grows with omitted-confounder strength.
>
> **On Q4: why CARE-SVD is not a manual top-k procedure.**
> **CARE-SVD is not a manual top-k pipeline**: the sparse plus low-rank fit induces the effective low-rank structure, and the practical question is only which recovered direction corresponds to \(Q\). By default we use the leading eigenvector, with human-anchor and loading-balance alternatives when needed.
>
> **On W5: CARE-Tensor sample-size sensitivity.**
> CARE-Tensor is unsupervised, so if more samples are needed they can come from additional unlabeled judge outputs (which is cheap) rather than new human labels (which is expensive). Empirically it is effective on moderate sample sizes: the smallest binary/preference dataset in the paper is SHP with 1,672 examples, where CARE-Tensor remains strong. We therefore view the theorem as a sufficient probabilistic guarantee, not a tight practical threshold.
>
> **On Q3 / W4: misspecification checks and empirical positioning against recent multi-judge baselines.**
> Exact misspecification bias cannot be easily measured on real benchmarks, because it is defined relative to an unobserved confounder-free quality latent. We instead address it through the misspecification theorem and controlled confounder experiments. **WS/UWS are relevant misspecified comparators** because they aggregate judges without explicitly separating latent quality from shared confounders. For recent LLM-specific baselines, PoLL is closest to AVG/MV, while PGED/GED and JudgeBlender target richer graph- or judge-ensemble settings whose behavior on our two-response benchmarks is largely represented by our current majority- and weighting-based baselines.
>
> **On Q5: adversarial robustness and why CARE-SVD is absent from that experiment.**
> The adversarial setup is binary, so CARE-Tensor is the main method there. We also have CARE-SVD results on the same experiment, showing improved robustness compared to MV and WS:
>
> | Attack | MV | WS | CARE-SVD | CARE-Tensor |
> |---|---:|---:|---:|---:|
> | `:` | 0.577 | 0.520 | **0.384** | 0.496 |
> | `,` | 0.441 | 0.546 | 0.417 | **0.000** |
> | `Solution` | 0.558 | 0.522 | 0.510 | **0.371** |
> | `Thought process:` | 0.674 | 0.587 | 0.494 | **0.000** |
> | `space` | 0.641 | 0.577 | 0.567 | **0.432** |
> | `Step by step` | 0.571 | 0.504 | 0.443 | **0.392** |
>
> **These results show that CARE-SVD is also more robust than MV/WS across all reported attacks, although CARE-Tensor is stronger overall in this binary setting.**
>
> **On W6: clarifying the CARE-Tensor presentation in the main text.**
> We agree that the CARE-Tensor path is more compressed in the main text. This was mainly a page-budget choice: both estimators instantiate the same confounder-aware framework, so we prioritized the shared model and theory in the main text and moved tensor-specific details to the appendix. In the revision we will add a short standalone CARE-Tensor summary in the main text so the method is understandable without heavy appendix lookup.

---

> > ### Author Rebuttal · Reviewer_RWpU · 2026-04-03
> >
> > Thanks for authors' response. I will keep the positive score here.

---

> > > ### Author Response · Authors · 2026-04-04
> > >
> > > We appreciate your constructive feedback which improved our overall paper quality and are glad that all your concerns are addressed.
> > >
> > > We greatly appreciate a reconsideration of the overall recommendation if all your questions were clarified.

---

### Official Review · Reviewer_Dn7a · 2026-03-12

**Soundness:** 3
**Presentation:** 3
**Significance:** 3
**Originality:** 2
**Overall Recommendation:** 4
**Confidence:** 3

**Summary:**

This paper introduces CARE, a confounder-aware aggregation framework for LLM-as-a-judge ensembles that addresses the problem of correlated errors among judges caused by shared latent confounders like verbosity bias or stylistic preferences. Standard aggregation methods such as majority voting and averaging implicitly assume judge independence, which fails in practice and can amplify systematic mistakes. CARE models judge scores as arising from both a true quality signal and shared confounding factors using a graphical model, then separates these through two complementary estimators: CARE-SVD, which uses sparse-plus-low-rank decomposition of the precision matrix for continuous Gaussian settings, and CARE-Tensor, which partitions judges into conditionally independent groups and applies tensor decomposition on third-order moments for discrete and mixture settings.

**Compliance With Llm Reviewing Policy:**

Affirmed.

**Ethical Review Concerns:**

N.A.

**Final Justification:**

The authors have addressed my concerns. Therefore, I would retain my rating of weak accept.

**Key Questions For Authors:**

When should practitioners choose CARE-SVD versus CARE-Tensor, and how does the method behave when its structural assumptions are violated?

The paper presents two estimators suited to different data regimes, but offers limited practical guidance on selection. CARE-SVD assumes quality is the dominant shared variation, which can fail when confounders are stronger, while CARE-Tensor requires a clean three-way partition of conditionally independent judges. In real deployments, practitioners may not know which regime applies or whether these structural assumptions hold. Could the authors provide diagnostic criteria or automated selection procedures, and characterize degradation more thoroughly when assumptions are partially violated on real data?

How does CARE scale to smaller judge panels, and what is the computational cost relative to simpler methods?

The experiments use 11–20 LLM judges, but many practical evaluation pipelines rely on only two or three models due to cost constraints. The paper does not report how performance degrades as the number of judges decreases, nor does it analyze the computational overhead of sparse-plus-low-rank decomposition and tensor decomposition compared to simple averaging or majority vote. Could the authors provide scaling experiments showing accuracy as a function of judge count, along with runtime comparisons, to help practitioners assess whether CARE's added complexity is justified in resource-constrained settings?

**Limitations:**

The limitations discussion is largely scattered across appendices rather than addressed centrally in the main text. The paper would benefit from a dedicated limitations section acknowledging key practical constraints, including the requirement for many judges, the reliance on validation labels despite the label free framing, the inconsistent gains across benchmarks, and the untested non Gaussian extensions. Being upfront about these in the main paper would give readers a more balanced picture.

On societal impact, the paper could go further in discussing the risk that improved aggregation methods may create a false sense of reliability, encouraging practitioners to trust automated evaluations in domains where human oversight remains essential. The framework's ability to suppress certain confounders could also inadvertently remove signals that are legitimately relevant to quality in some contexts. For instance, readability or formatting may genuinely matter for certain evaluation tasks, and treating them as confounders could systematically disadvantage certain response styles.

**Strengths And Weaknesses:**

Strengths:

The theoretical foundations are rigorous, with formal identifiability guarantees (Propositions 4.1, Theorems 4.2–4.3) and finite-sample recovery bounds for both the SVD and tensor paths.

The paper is well-structured, with the three core challenges clearly laid out early and then systematically addressed.

The paper tackles a real and increasingly important problem as LLM-as-a-judge pipelines become standard practice in evaluation and RLHF.

Weaknesses:

The theoretical guarantees depend on idealized assumptions such as orthonormal latent-observable connections and diagonal latent precision that are difficult to verify with real LLM judges. The stability results for approximate orthogonality help but do not fully close the gap between theory and practice.

The symmetry-breaking heuristic selects the leading eigenvector as the quality factor, assuming quality drives the dominant shared variation. This assumption can fail precisely when a confounder like verbosity is stronger than the quality signal, which is the core scenario the paper aims to address.

CARE-Tensor requires partitioning judges into exactly three conditionally independent groups, but the paper does not analyze sensitivity when no clean partition exists. In practice, LLM judges may have overlapping dependencies that resist clean grouping.

Theoretical quantities like the tangent space curvature constant $\xi(T)$ appear in the bounds but are difficult for practitioners to interpret or compute, limiting the practical utility of the sample complexity results.

The method requires 11–20 LLM judges, which significantly increases evaluation cost and may not reflect typical deployment scenarios where practitioners use only a small handful of judges. The paper does not discuss computational overhead or scalability tradeoffs.

Performance gains are inconsistent across benchmarks, e.g., CARE-SVD underperforms simpler baselines like MACE or AVG on several classification datasets in Table 2, and some improvements are within a few percentage points, raising questions about when the added complexity is justified.

The core technical components, that is, sparse-plus-low-rank decomposition from Chandrasekaran et al. (2012) and tensor decomposition from Anandkumar et al. (2014), are well-established, and the contribution is primarily in combining and applying them to the LLM evaluation setting rather than developing new machinery.

---

> ### Author Rebuttal · Authors · 2026-03-31
>
> We thank the reviewer for recognizing the strength of the theoretical development, the overall structure of the paper, and the importance of the problem setting. We address below the main concerns about estimator choice, assumption violations, novelty, and operating regime.
>
> **On when to use CARE-SVD vs. CARE-Tensor, and what happens when assumptions fail.**
> We provide the main selection rule: CARE-SVD is the natural choice for continuous, approximately Gaussian scores, whereas CARE-Tensor is designed for discrete / preference settings. CARE-SVD can still be applied in discrete settings, but there it is more weakly identified and typically less reliable. The theory-side assumptions are identifiability conditions rather than claims that real judge panels exactly satisfy a directly testable model. We go beyond exact recovery: Section 3 gives stability under approximate orthogonality, and Appendix D.4 provides an explicit misspecification bound.
>
> **On the leading-eigenvector heuristic.**
> The leading-eigenvector rule is a default symmetry-breaking heuristic, not a claim that quality always drives the dominant shared variation. When the dominant factor is confounding, our draft discusses anchor-based identification, loading-balance heuristics, and non-orthogonality correction.
>
> **On the three-group assumption in CARE-Tensor.**
> CARE-Tensor likewise does not assume a hand-specified partition: it first learns judge dependencies and then searches for a three-way grouping with low cross-group dependence. Appendix E studies violated grouping assumptions. Random grouping degrades recovery substantially, while graph-aware grouping reduces reconstruction error by more than an order of magnitude.
>
> **On computational cost.**
> **CARE adds a one-time offline fitting step, while deployment-time inference remains lightweight.** CARE-SVD reduces to applying learned weights at inference time, and CARE-Tensor applies the fitted latent-state predictor, so both act as post-hoc aggregation steps on fixed judge outputs. On SHP, the per-example inference times are:
>
> | Method | Infer (ms/ex.) |
> |---|---:|
> | AVG | 0.000306 |
> | MV | 0.000207 |
> | WS | 0.011064 |
> | UWS | 0.000008 |
> | CARE-SVD | 0.000010 |
> | CARE-Tensor | 0.011226 |
>
> Fitting on SHP takes under 1s for CARE-SVD and about 1 minute for CARE-Tensor, but this is a one-time offline cost rather than a per-example deployment cost.
>
> **On smaller judge panels.**
> Thank you for this observation! **CARE does not require 11--20 judges**; those simply happened to be the panel sizes in our current benchmarks. To emphasize this point, we ran new 10-seed random-subset ablations and summarize CARE by relative improvement over the best baseline in all datasets:
>
> | Judges | CARE-Tensor | CARE-SVD |
> |---|---:|---:|
> | 3 | +4.1\% acc. | -5.4\% MAE |
> | 5 | +7.2\% acc. | -0.2\% MAE |
> | 8 | +9.3\% acc. | +2.1\% MAE |
> | 11 | +5.6\% acc. | +2.7\% MAE |
>
> CARE-Tensor delivers consistent gains even in smaller panels, while CARE-SVD improves more gradually and becomes beneficial as the panel grows. This is plausible because CARE-Tensor works with a low-dimensional discrete latent state space, so a few informative judge groups can separate the latent states reasonably well. CARE-SVD, by contrast, must recover and disentangle continuous spectral directions from covariance structure, which is more sensitive to noise and weak eigengaps in smaller panels.
>
> **On novelty relative to prior sparse+low-rank and tensor methods.**
> The decomposition tools themselves are classical, and we do not claim to invent a new generic sparse+low-rank or tensor algorithm. **Our contribution is the *confounder-aware formulation* of multi-judge LLM aggregation**, together with concrete estimator instantiations for continuous and preference settings, identifiability and misspecification analysis specialized to this setting, and practical procedures for quality-factor identification and graph-aware view formation. Importantly, CARE-Tensor is not a standalone off-the-shelf tensor application: it combines tensor decomposition with sparse+low-rank structure estimation, using the judge graph recovered by the S+L step to form approximate multi-view partitions. This yields a better-conditioned setting in which the tensor assumptions are more plausible and recovery is more stable. In that sense, the novelty lies not in proposing a new decomposition primitive, but in turning these ingredients into a unified framework for LLM-judge aggregation under shared latent confounders.

---

> > ### Author Rebuttal · Reviewer_Dn7a · 2026-04-03
> >
> > The authors have addressed my concerns.

---

> > > ### Author Response · Authors · 2026-04-04
> > >
> > > Thank you for positive feedback. We are glad that all your concerns are addressed and it improved our overall paper.
> > >
> > > If all of your questions were clarified, we greatly appreciate a reconsideration of the overall recommendation.

---

### Official Review · Reviewer_i25w · 2026-03-16

**Soundness:** 3
**Presentation:** 3
**Significance:** 3
**Originality:** 2
**Overall Recommendation:** 5
**Confidence:** 3

**Summary:**

This paper studies aggregation of multiple LLM judges under shared latent confounders. The main idea is that standard aggregation methods, such as majority voting or averaging, implicitly assume judge independence, while in practice judges often make correlated errors due to factors such as verbosity, style, or shared training artifacts.

**Compliance With Llm Reviewing Policy:**

Affirmed.

**Final Justification:**

The rebuttal has addressed all of my concerns. I support the acceptance of this paper.

**Key Questions For Authors:**

I want to confirm my following statement is correct or not:

1. The experimental section is solid, but the baseline set is still incomplete, especially with respect to more recent LLM-specific judge aggregation or debiasing methods.

2. The paper gives limited practical guidance on when to use CARE-SVD versus CARE-Tensor, how to choose the rank, and how to diagnose whether shared confounders are actually present.

**Limitations:**

The main weakness is the gap between theory and practice.

L1. The theoretical results rely on fairly strong structural assumptions, but the paper does not sufficiently explain when these assumptions are realistic in real judge panels or how performance degrades when they fail.

L2. The paper repeatedly emphasizes operation without ground-truth labels, but the practical pipeline still relies on symmetry-breaking heuristics and validation choices, so this claim should be stated more carefully.

**Strengths And Weaknesses:**

First, the paper addresses an important and timely problem in LLM-as-a-judge evaluation. The motivation is clear and practically relevant.

Second, the method is not just a heuristic reweighting scheme; it introduces a more principled latent-variable view of judge aggregation.

Third, the paper includes meaningful theoretical analysis and fairly broad experiments across scoring, classification, and preference settings.

---

> ### Author Rebuttal · Authors · 2026-03-31
>
> We thank the reviewer for the thoughtful comments and for recognizing the importance of the problem and the strength of the experimental section. We address each point below.
>
> **On the baseline set and recent LLM-specific aggregation / debiasing methods.**
> Although our baselines use classical names, they cover most recent LLM-specific methods relevant to our setting. Recent work falls into two groups. On the *group / panel* side, PoLL [1], PGED/GED [2], and JudgeBlender [3] aggregate multiple judges or prompts. On the *individual-judge* side, JudgeLM [4], PORTIA [5], and CalibraEval [6] improve or calibrate a single evaluator. Only the first group is directly comparable to CARE, and we will map these connections explicitly in the revised Related Work section.
>
> Within that directly comparable group, PoLL is closest to our panel-style `AVG` / `MV` baselines. PGED/GED is aimed at richer multi-candidate preference graphs, but on our two-response A-vs.-B benchmarks it largely reduces to pairwise majority (`MV`) or weighted-majority aggregation (`WS`). Thus, on our current benchmarks, these recent LLM-specific group-level methods are reasonably reflected by our existing baselines.
>
> **On practical guidance for CARE-SVD vs. CARE-Tensor, rank, and confounder diagnosis.**
> The paper provides much of this guidance, though parts were previously split across the method and appendix. We have unified it in the main body for clarity. CARE-SVD is the natural estimator for continuous settings, whereas CARE-Tensor is designed for discrete / preference settings. CARE-SVD can still be applied in discrete settings, but identifiability is weaker there, making it typically less reliable.
>
> Latent rank is also not a standalone user-chosen hyperparameter in our current pipeline. For CARE-SVD, the effective rank is induced by the sparse+low-rank fit and governed by the regularization hyperparameters; in our current CARE-Tensor setup, the latent-state structure is fixed by the binary $(Q,C)$ model. For confounder diagnosis, we use factor-loading interpretation (Section 5.2), anchor / loading-balance heuristics, and controlled confounder analyses to distinguish broad quality factors from more concentrated confounders.
>
> **On the realism of the theoretical assumptions and how violations affect performance.**
> The assumptions in our theory are intended to capture the regime CARE targets: a small number of shared latent factors (true quality plus broad confounders such as verbosity or style) together with sparse residual judge-judge dependencies. This is a natural approximation for multi-judge LLM evaluation, where judges are neither independent nor driven by a single factor.
>
> The paper **analyzes how performance degrades when assumptions fail**. For CARE-SVD, Section 3 proves stability under approximate orthogonality, and Appendix D.4 gives an explicit misspecification bound showing that the error increases with omitted-confounder strength and decreases with quality-signal strength. For CARE-Tensor, Appendix E shows that violating the three-view independence assumption via random partitioning degrades recovery, while graph-aware partitioning restores performance. Thus, the degradation story is explicit in both theory and experiments, and we will further emphasize this in the updated draft.
>
> **On the claim of operating without ground-truth labels.**
> The intended claim is that CARE learns from observed judge scores $J$ without directly observing the latent ground-truth quality variable $Q$. Any instantiation of CARE still requires a symmetry-breaking step for latent-variable identifiability, namely to determine which recovered latent corresponds to $Q$ rather than a confounder. In the paper, this step is handled by unsupervised heuristics by default, while ground-truth samples or anchor sets are explicitly optional.
>
> Similarly, the validation choices in our experiments are used for hyperparameter tuning and judge selection, not for directly identifying the latent true-quality factor $Q$ or learning its aggregation weights. We will state this more carefully in the revision: **CARE does not require direct observation** of the latent true-quality variable for model learning and aggregation.
>
> **References**
>
> [1] Pat Verga et al. *Replacing Judges with Juries: Evaluating LLM Generations with a Panel of Diverse Models*. arXiv, 2024.
>
> [2] Zhengyu Hu et al. *Towards Acyclic Preference Evaluation of Language Models via Multiple Evaluators*. AAAI, 2026.
>
> [3] Hossein A. Rahmani et al. *JudgeBlender: Ensembling Judgments for Automatic Relevance Assessment*. WWW, 2025.
>
> [4] Lianghui Zhu et al. *JudgeLM: Fine-tuned Large Language Models are Scalable Judges*. ICLR, 2025.
>
> [5] Zongjie Li et al. *Split and Merge: Aligning Position Biases in LLM-based Evaluators*. EMNLP, 2024.
>
> [6] Haitao Li et al. *CalibraEval: Calibrating Prediction Distribution to Mitigate Selection Bias in LLMs-as-Judges*. ACL, 2025.

---

> > ### Author Rebuttal · Reviewer_i25w · 2026-04-04
> >
> > Thank you for the thorough rebuttal. Moreover, in the context of aggregation, one recent study (https://openreview.net/forum?id=XNbVoi9mfr) suggests that rankings are more reliable than ratings and that aggregate rankings are a somewhat debiasing approach. I am unsure how this perspective is received in the relevant literature, perhaps in social choice theory, but I believe it would be valuable for the discussion of related work.

---

> > > ### Author Response · Authors · 2026-04-04
> > >
> > > Thank you for your positive response. We appreciate your constructive feedback, which has helped improve the clarity and overall quality of the paper.
> > >
> > > We also thank you for pointing out this relevant work. It studies ranking-based aggregation, which is closely related to our setting. Our framework models judge outputs through latent quality and shared confounders, and can be naturally extended to ranking-based aggregation. Exploring this in practice is an interesting direction for future work. We will include this work in the related work and discuss these connections.
> > >
> > > If your concerns have been adequately addressed, we would greatly appreciate reconsideration of the overall recommendation.

---

### Decision · Program_Chairs · 2026-04-30

**Decision:**

Accept (regular)

**Comment:**

All reviewers are positive. The concerns raised in the initial reviews have been fully addressed in the rebuttal. Hence, I recommend accepting the paper.